# The interplay between homeostatic synaptic scaling and homeostatic structural plasticity maintains the robust firing rate of neural networks

Han Lu[1,2,3]*[†], Sandra Diaz-Pier[3], Maximilian Lenz[1,4], Andreas Vlachos[1,2,5]*

[1]Department of Neuroanatomy, Institute of Anatomy and Cell Biology, Faculty of Medicine, University of Freiburg, Freiburg, Germany; [2]Center BrainLinks-BrainTools, University of Freiburg, Freiburg, Germany; [3]Simulation and Data Lab Neuroscience, Jülich Supercomputing Centre (JSC), Institute for Advanced Simulation, Forschungszentrum Jülich GmbH, Jülich, Germany; [4]Institute of Neuroanatomy and Cell Biology, Hannover Medical School, Hannover, Germany; [5]Center for Basics in Neuromodulation (NeuroModulBasics), Faculty of Medicine, University of Freiburg, Freiburg, Germany

*For correspondence:
ha.lu@fz-juelich.de (HL);
andreas.vlachos@anat.uni-freiburg.de (AV)

Present address: [†]Simulation and Data Lab Neuroscience, Jülich Supercomputing Centre (JSC), Institute for Advanced Simulation, Forschungszentrum Jülich GmbH, Jülich, Germany

Competing interest: The authors declare that no competing interests exist.

## eLife Assessment

This **valuable** study combines experiments and modelling to advance our understanding of the nonlinear nature of homeostatic structural plasticity and its interaction with synaptic scaling. The methodology and findings are **solid**, although additional work is needed to better link models with experiments and support some of the conclusions drawn. This study will be of interest to theoretical and experimental neuroscientists working in homeostatic plasticity.

**Abstract** Critical network states and neural plasticity enable adaptive behavior in dynamic environments, supporting efficient information processing and experience-dependent learning. Synaptic-weight-based Hebbian plasticity and homeostatic synaptic scaling are key mechanisms that enable memory while stabilizing network dynamics. However, the role of structural plasticity as a homeostatic mechanism remains less consistently reported, particularly under activity inhibition, leading to an incomplete understanding of its functional impact. In this study, we combined live-cell microscopy of eGFP-labeled neurons in mouse organotypic entorhinal-hippocampal tissue cultures (Thy1-eGFP mice of both sexes) with computational modeling to investigate how synapse-number-based structural plasticity responds to activity perturbations and interacts with homeostatic synaptic scaling. Tracking individual dendritic segments, we found that inhibiting excitatory neurotransmission does not monotonically regulate dendritic spine density. Specifically, inhibition of AMPA receptors with 200 nM 2,3-dioxo-6-nitro-7-sulfamoyl-benzo[f]quinoxaline (NBQX) increased spine density, whereas complete AMPA receptor blockade with 50 µM NBQX reduced it. Motivated by these findings, we developed network simulations incorporating a biphasic structural plasticity rule governing activity-dependent synapse formation. These simulations showed that the biphasic rule maintains neural activity homeostasis under stimulation and permits either synapse formation or synapse loss depending on the degree of activity deprivation. Homeostatic synaptic scaling further modulated recurrent connectivity, network activity, and structural plasticity outcomes. It reduced stimulation-triggered synapse loss by downscaling synaptic weights and rescued silencing-induced synapse loss by upscaling recurrent input, thus reactivating silent neurons. The interaction between

these mechanisms provides a mechanistic explanation for divergent findings in the literature. In summary, homeostatic synaptic scaling and homeostatic structural plasticity dynamically compete and compensate for each other, ensuring efficient and robust control of firing rate homeostasis.

## Introduction

To survive in dynamic environments, animals must quickly respond to familiar cues, such as those signaling food or predators, while remaining alert to novel stimuli. The former reflects experience-dependent learning, where subtle memory cues can trigger strong responses in the corresponding pathway during recall (*Cabeza et al., 1997*; *Nakazawa et al., 2002*; *Wade-Bohleber et al., 2019*; *Bone et al., 2020*). The latter requires brain networks to maintain activity in a critical state, allowing efficient information transmission via action potentials and neural avalanches (*Beggs and Plenz, 2003*; *Beggs and Plenz, 2004*; *Petermann et al., 2009*; *Shew et al., 2009*; *Hahn et al., 2010*; *Shew and Plenz, 2013*). These processes represent two seemingly opposing but well-coordinated mechanisms: associative learning and firing rate homeostasis.

The concept of firing rate homeostasis has been hovering in theories since the discovery of long-term synaptic potentiation (LTP; *Bliss and Lømo, 1973*). LTP is a positive feedback mechanism that adjusts synaptic strength among excitatory neurons ('neurons that fire together, wire together', *Löwel and Singer, 1992*), as postulated by *Hebb, 1949*. While its associative properties allow synapses to preserve memory traces by increasing synaptic weights, it also risks driving network dynamics toward overexcitation or silence (*Bliss and Collingridge, 1993*; *Figure 1Ai, ii, Bi*). *In vivo* studies in rodents demonstrated firing rate homeostasis in the visual cortex and hippocampus, where neural activity is restored at both the single-neuron (*Hengen et al., 2016*; *Ma et al., 2019*; *Torrado Pacheco et al., 2019*) and population level (*Slomowitz et al., 2015*; *Lu et al., 2019*), within days of perturbation. This ability to stabilize network dynamics without erasing memories underscores the robustness of the brain's complex networks (*Alon et al., 1999*; *von Dassow et al., 2000*; *Carlson and Doyle, 2002*; *Kitano, 2002*; *Kitano, 2004*; *Aoki et al., 2019*).

Robustness is a fundamental property of complex biological and engineering systems, enabling adaptation to external perturbations by restoring an original attractor state or shifting to a new one while preserving function (*Kitano, 2002*; *Kitano, 2004*). Key features of robustness include negative feedback control, redundancy, and heterogeneity (*Alon et al., 1999*; *von Dassow et al., 2000*; *Carlson and Doyle, 2002*; *Kitano, 2002*; *Kitano, 2004*; *Aoki et al., 2019*). For instance, homeostatic synaptic scaling exemplifies a negative feedback mechanism (*Turrigiano et al., 1998*; *Figure 1Aiii, Bii*), where neurons proportionally adjust their synaptic weights to compensate for long-term changes in activity (*Turrigiano et al., 1998*), membrane potential (*Leslie et al., 2001*), or intracellular calcium concentration (*Slutsky et al., 2004*). Redundancy and heterogeneity describe the presence of multiple mechanisms capable of achieving the same outcome, ensuring system robustness against failure. Although redundancy and heterogeneity have been recognized in biological systems in the form of genetic buffering and convergent molecular circuits (*Csete and Doyle, 2002*; *Kitano, 2002*; *Kitano, 2004*), their representation in activity-dependent plasticity has been less explicitly discussed. Mechanisms such as homosynaptic long-term depression (LTD; *Lynch et al., 1977*) under the Bienenstock–Cooper–Munro (BCM) rule (*Bienenstock et al., 1982*) or spike-timing-dependent plasticity (STDP; *Markram et al., 1997*; *Bi and Poo, 1998*; *Gütig et al., 2003*; *Izhikevich and Desai, 2003*), and heterosynaptic LTD via synaptic tagging and capture (*Redondo and Morris, 2011*) help stabilize firing rates. Inhibitory plasticity also restores the excitation-inhibition balance (*Froemke, 2015*; *Hennequin et al., 2017*; *Hiratani and Fukai, 2017*). These mechanisms, whether modulating synaptic transmission locally or globally, demonstrate functional redundancy and implementational heterogeneity, aligning with homeostatic synaptic scaling to maintain network stability.

However, it remains unclear whether structural plasticity exhibits similar redundancy. Structural plasticity encompasses changes in dendritic spine sizes and numbers, axonal boutons, synapse count, and network connectivity——all critical for synaptic transmission and memory capacity (*Emina and Kropff, 2022*). The homeostatic structural plasticity (HSP) model, which posits that synapse formation and elimination are homeostatically regulated (*Butz et al., 2009a*; *Butz et al., 2009b*; *Butz and van Ooyen, 2013*; *Butz et al., 2014*; *Diaz Pier et al., 2016*), has expanded theoretical models of network reorganization following diverse perturbations (*Gallinaro and Rotter, 2018*; *Lu et al., 2019*; *Gallinaro*

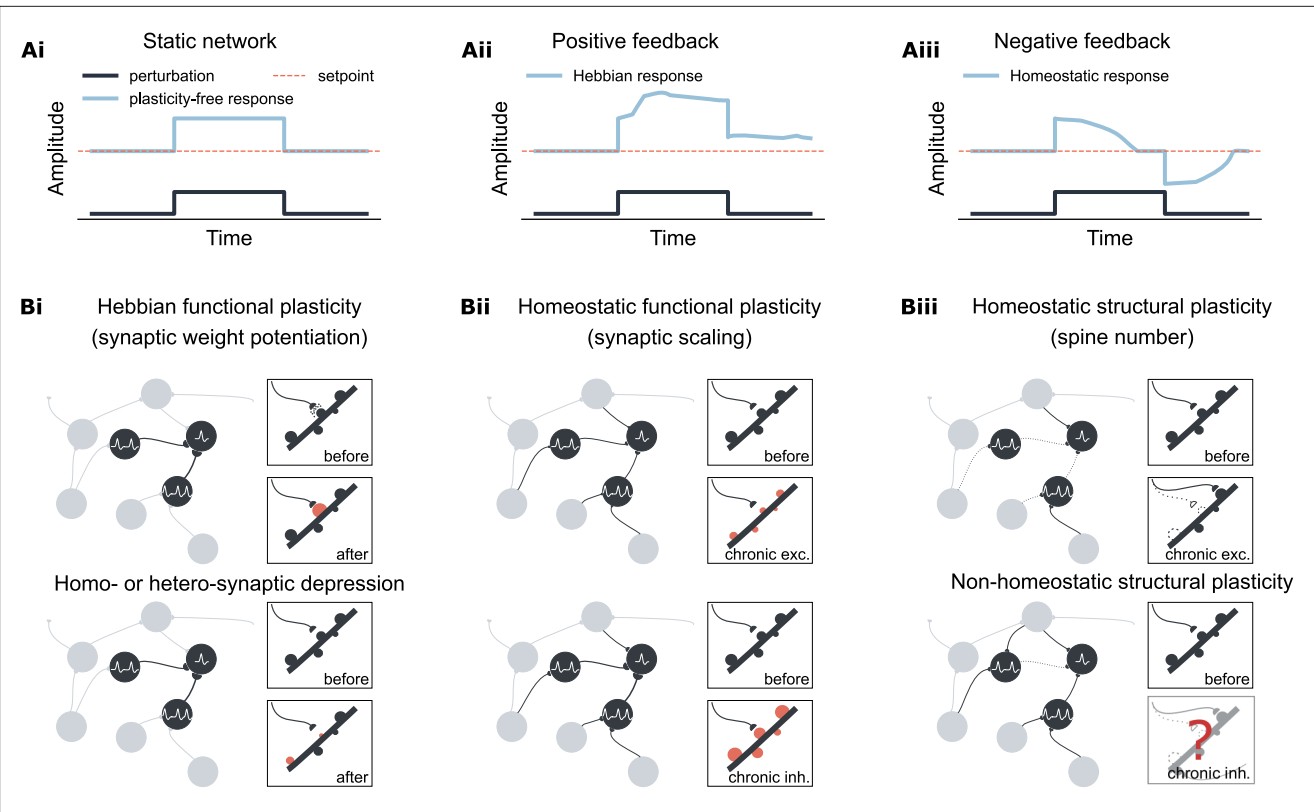

**Figure 1.** Overview of Hebbian and homeostatic plasticity. (**Ai**) In a static network, neural activity is driven by external input. (**Aii**) Hebbian plasticity amplifies network response via a positive feedback mechanism, increasing sensitivity to external input. (**Aiii**) Homeostatic plasticity restores activity toward a setpoint through negative feedback, supporting firing rate homeostasis. (**Bi**) Hebbian plasticity is synapse-specific, strengthening recurrent connectivity by potentiating the weights of activated synapses. In contrast, synaptic depression can occur either in the same synapse (homosynaptic depression) or in neighboring synapses (heterosynaptic depression) depending on the stimulation protocol. (**Bii**) Homeostatic synaptic scaling is a cell-autonomous process involving proportional up- or down-scaling of all input synaptic weights in response to prolonged changes in postsynaptic activity. (**Biii**) Homeostatic structural plasticity is likewise cell-autonomous and involves compensatory spine loss during chronic excitation. In contrast, chronic inhibition can lead to divergent and sometimes non-homeostatic changes in spine density.

*et al., 2022*; *Lu et al., 2022*; *Anil et al., 2023*). Nonetheless, experimental findings present a more complex picture (*Figure 1Biii*): synaptic downscaling is often associated with spine loss, while chronic activity deprivation leads to inconsistent and mostly non-homeostatic spine loss (*Moulin et al., 2022*). This complexity highlights the need for a deeper understanding of activity-dependent structural plasticity, particularly the rules governing spine number changes. A potentially overlooked factor is the degree of inhibition. Given the variability in experimental conditions—ranging from monocular deprivation to lesions and pharmacological treatments—we hypothesized that: (i) The relationship between neural activity and spine numbers is non-monotonic, leading to divergent responses under different levels of inhibition. (ii) Structural responses may be influenced by interactions with other plasticity mechanisms, especially homeostatic synaptic scaling, which is crucial during chronic activity deprivation, as evidenced by both computational models and experimental data.

In this study, we applied two concentrations of the competitive AMPA-receptor antagonist NBQX to gradually suppress neural activity. Using whole-cell patch-clamp recordings and time-lapse imaging, we found that activity deprivation affects dendritic spine numbers and sizes in a non-monotonic manner: a low concentration of NBQX (200 nM) increased spine numbers, while a higher concentration (50 μM) reduced them. These findings guided the development of a spiking neural network model with a biphasic, synapse-number-based HSP rule, which successfully reproduced both homeostatic and non-homeostatic spine changes under different levels of activity perturbations. Additionally, we implemented a monotonic, synaptic-weight-based homeostatic synaptic scaling (HSS) rule, as described in the literature (*van Rossum et al., 2000*). While largely redundant to the biphasic

HSP rule, the HSS rule facilitates the transition from non-homeostatic synapse loss to homeostatic synapse regeneration by adjusting recurrent connectivity in silenced networks. Our results highlight the redundancy and complementarity between HSP and HSS rules, underscoring their combined role in ensuring robust and efficient adaptation of network activity (*Yi et al., 2000*; *Briat et al., 2016*; *Aoki et al., 2019*).

## Results

### Experimental evidence supports a non-monotonic structural plasticity rule under graded activity blockade

Variability in experimental results regarding structural plasticity under chronic activity deprivation (*Moulin et al., 2022*) may arise from (i) non-monotonic, activity-dependent regulation of spine numbers, (ii) concurrent changes in synaptic weights and firing rates due to synaptic scaling, or (iii) a

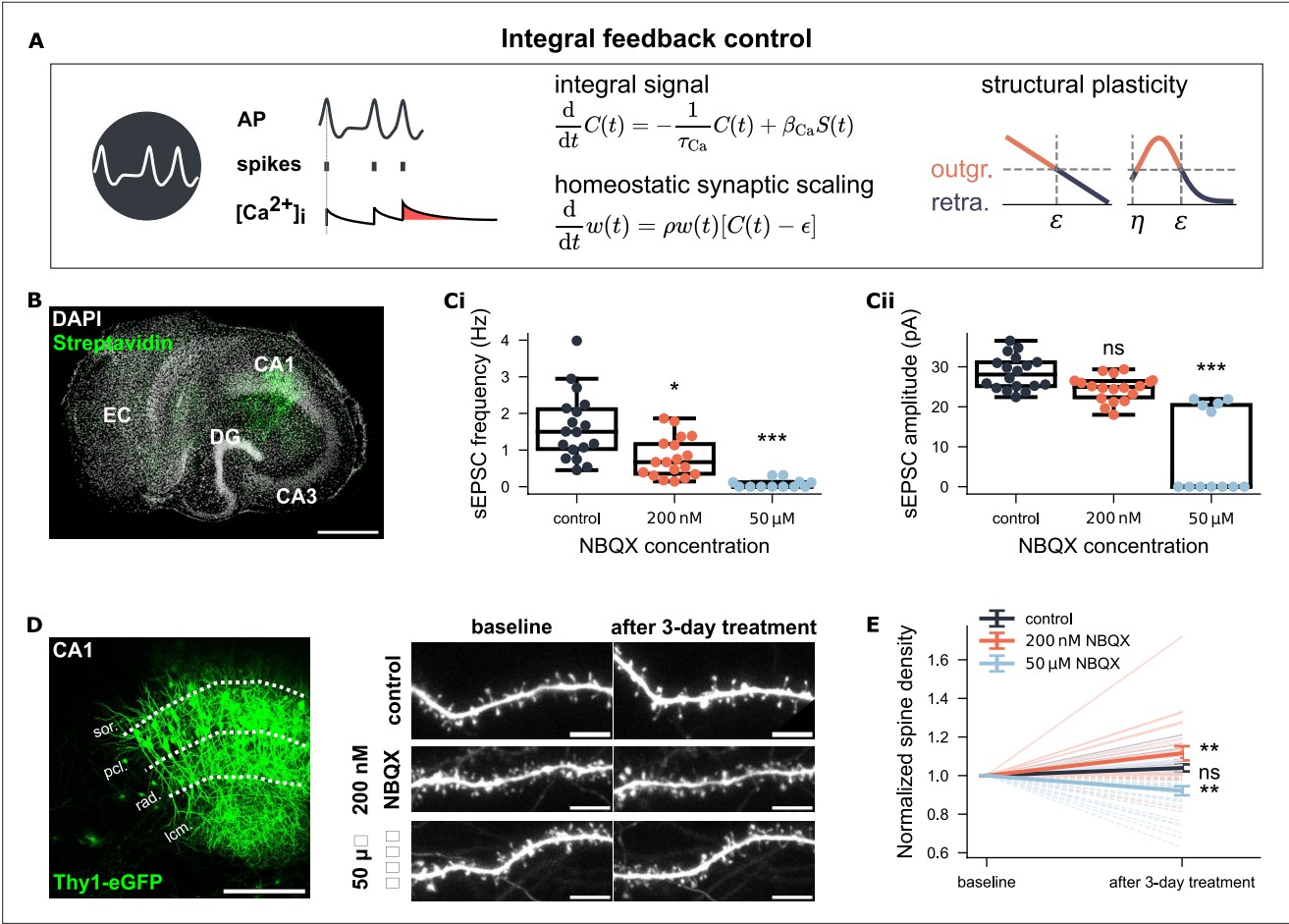

**Figure 2.** Non-monotonic spine density changes under NBQX inhibition support a biphasic structural plasticity rule. (**A**) Integral feedback control framework for homeostatic synaptic scaling and structural plasticity. Both mechanisms use intracellular calcium concentration ([Ca$^{2+}$]$_i$, $C(t)$) to track neural activity (AP, action potential, $S(t)$). Calcium increases with each postsynaptic spike via calcium influx ($\beta_{Ca}$) and decays with a time constant $\tau_{Ca}$. Synaptic scaling adjusts synaptic weights $w(t)$ multiplicatively using a scaling factor $\rho$, based on the deviation of calcium from a setpoint $\epsilon$. Structural plasticity similarly uses $C(t)$ to regulate the growth and retraction of axonal boutons and dendritic spines. Two variants of the structural plasticity rule are illustrated: one with a linear and the other with a non-linear (biphasic) calcium dependency. (**B**) Example CA1 pyramidal neuron recorded and *post hoc* identified in an entorhinal-hippocampal tissue culture. Scale bar: 500 μm. (**Ci-Cii**) Group data of AMPA receptor-mediated sEPSC from three experimental groups (control: N=18; 200 nM NBQX: N=18; 50 μM NBQX: N=12). (**D**) Representative Thy1-eGFP culture and time-lapse imaging of dendritic segments in the stratum radiatum (rad.) before and after three-day treatment. Scale bars: 200 μm, 5 μm. (**E**) Spine density at baseline and after treatment. Values are normalized to the respective baseline per segment. Light-shaded lines show individual trajectories (solid = increase; dashed = decrease). Dark-shaded lines with error bars represent group means and SEM for each group (control: N=19; 200 nM NBQX: N=24; 50 μM NBQX: N=33).

combination of both. To systematically address this, we designed computer simulations that incorporate a synapse-number-based structural plasticity rule alone and in combination with synaptic-weight-based synaptic scaling to examine their interactions. In our implementation, these activity-dependent models act as integral feedback mechanisms, both governed by intracellular calcium concentration $C(t)$ and a setpoint value $\epsilon$ (*Figure 2A*), which monitor neural activity and influence synaptic weights or synapse numbers. This approach eliminates the need for manual time-scale alignment. However, structural plasticity models have different variants—some proposing that neural activity influences neurite growth and retraction in a monotonic way (*Gallinaro and Rotter, 2018*; *Lu et al., 2019*; *Gallinaro et al., 2022*) or some in a non-monotonic manner (*Butz et al., 2009a*; *Butz and van Ooyen, 2013*; *Butz et al., 2014*; *Diaz Pier et al., 2016*). This highlights the need for additional experimental data to refine model selection.

To test whether spine density responds monotonically to deprivation in neural activity, we used two concentrations of the AMPA receptor antagonist NBQX, 200 nM and 50 μM, and studied CA1 pyramidal neurons in mouse organotypic entorhinal-hippocampal tissue cultures (*Figure 2B*). Whole-cell patch-clamp recording of AMPA receptor-mediated spontaneous excitatory postsynaptic currents (sEPSCs) revealed a significant reduction in mean sEPSC amplitude with 200 nM NBQX and a near-to-complete blockade of excitatory synaptic transmission with 50 μM NBQX (*Figure 2Ci–Cii*). Both NBQX

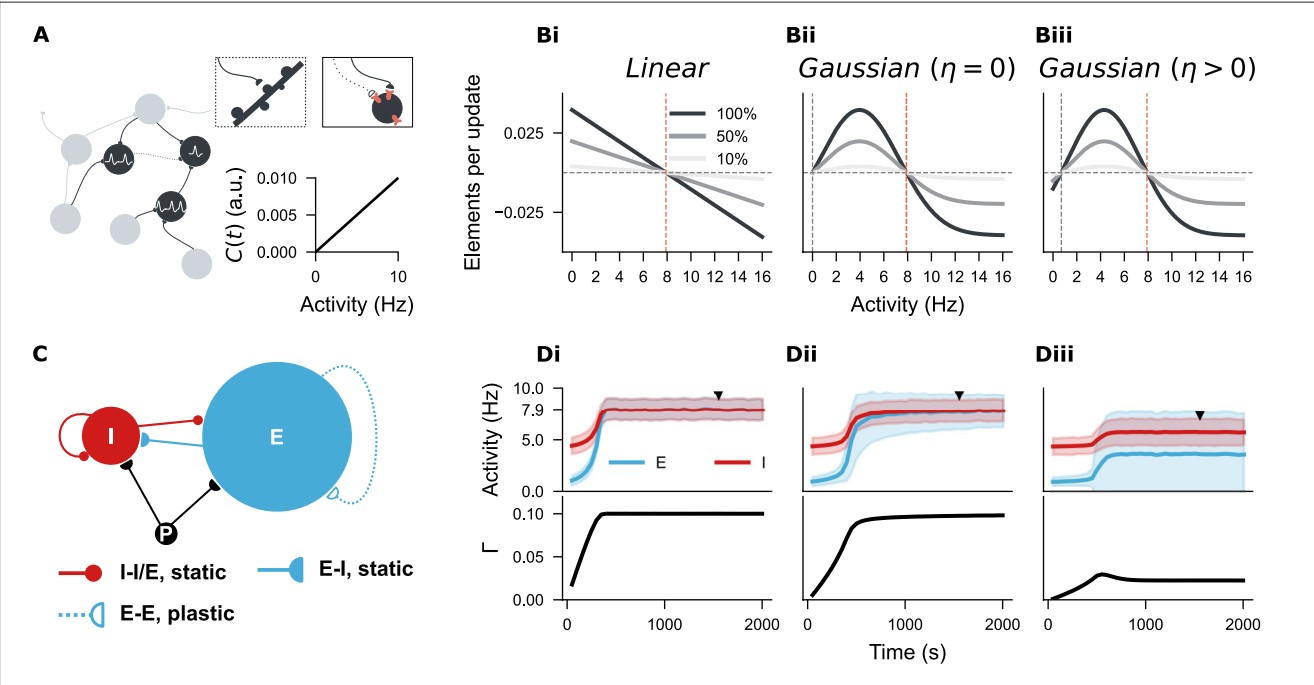

**Figure 3.** Establishing a spiking neural network with three distinct structural plasticity rules. (**A**) A network of point neurons was used to study structural plasticity. Dendritic spines are visualized as pink sticks on the soma, and axonal boutons are represented by half-circles——solid or empty depending on their growth state. An empty half-circle with a dashed line denotes a retracting axon. In this model, intracellular calcium concentration $C(t)$ is linearly correlated with neural firing rate, so neural activity is used throughout this study as a proxy for 'firing rate homeostasis', given its physical unit compared to the arbitrary unit (a.u.) of calcium concentration. (**Bi-Biii**) Three structural plasticity growth rules regulate changes in synaptic element numbers. (**Bi**) Linear rule with a single setpoint ($\epsilon = 7.9$, orange line). (**Bii**) Gaussian rule with two setpoints, one at zero ($\eta = 0$, grey line; $\epsilon = 7.9$, orange line). (**Biii**) Gaussian rule with two non-zero setpoints ($\eta = 0.7$, grey line; $\epsilon = 7.9$, orange line). Three shades represent 100%, 50%, or 10% of the original growth rate ($\nu$), with positive and negative values indicating the rate of synaptic element growth and loss, respectively. (**C**) Network architecture based on the Brunel network model, consisting of 10000 excitatory (E, blue) and 2500 inhibitory neurons (I, red), receiving external Poisson input (P). Synapses among I-I, I-E, and E-I populations are hard-wired with a fixed 10% probability. E-E synapses are subject to structural plasticity rules. (**Di-Diii**) Temporal evolution of neural activity and network connectivity ($\Gamma$) during network growth, guided by the three plasticity rules. Unless otherwise stated, the curves and shaded areas represent the mean and standard deviation of firing rates in E and I populations. An equilibrium state ($\Gamma = 10\%$) was reached under the linear and zero-Gaussian rules (**Di, Dii**), but not with the biphasic Gaussian rule (**Diii**). Firing rate distributions and network connectivity matrices at the marked time points (solid triangles) are shown in *Figure 3—figure supplement 1* for **Di-Dii**, and in *Figure 4* for **Diii**.

The online version of this article includes the following figure supplement(s) for figure 3:

**Figure supplement 1.** Firing rate distributions and connectivity matrices at the corresponding time points shown in *Figure 3*.

concentrations were applied to Thy1-eGFP cultures for three days to induce chronic activity inhibition. Individual dendritic segments (side branches of apical dendrites in the stratum radiatum) were imaged before and after treatment using time-lapse microscopy (*Figure 2D*). The analysis showed that 200 nM NBQX increased spine density ($p = 0.003$, Wilcoxon test), while 50 μM NBQX reduced spine density ($p = 0.008$, Wilcoxon test), compared to baseline (*Figure 2E*). No significant changes in spine density were observed in control cultures ($p = 0.06$, Wilcoxon test). These results demonstrated a non-monotonic relationship between neural activity and spine numbers under activity deprivation: partial inhibition increased spine numbers, whereas complete inhibition decreased them.

## Stabilizing and characterizing a biphasic synapse-number-based structural plasticity rule

Since changes in spine numbers or density are functionally linked to synapse formation, loss, and rewiring, we aimed to establish a synapse-number-based structural plasticity rule in point neurons and study synaptic rewiring in a large, topology-free spiking neuron network to examine the first hypothesis, thereby bypassing the complexities of neuronal morphology (*Figure 3A*). For comparison, we first simulated the classical linear growth rule (*Figure 3Bi*). To capture the observed non-monotonic dependency in a simplified manner, we adopted a Gaussian-shaped growth rule for synaptic element numbers with two setpoints. By setting the first setpoint at either $\eta = 0$ or $\eta > 0$, we generated two variants of the Gaussian rule (*Figure 3Bii–Biii*). We then grew three neural networks in which synapses among excitatory neurons were governed by these three rules, respectively (*Figure 3C*). As reported in previous studies (*Diaz Pier et al., 2016*; *Gallinaro and Rotter, 2018*; *Lu et al., 2019*; *Lu et al.,*

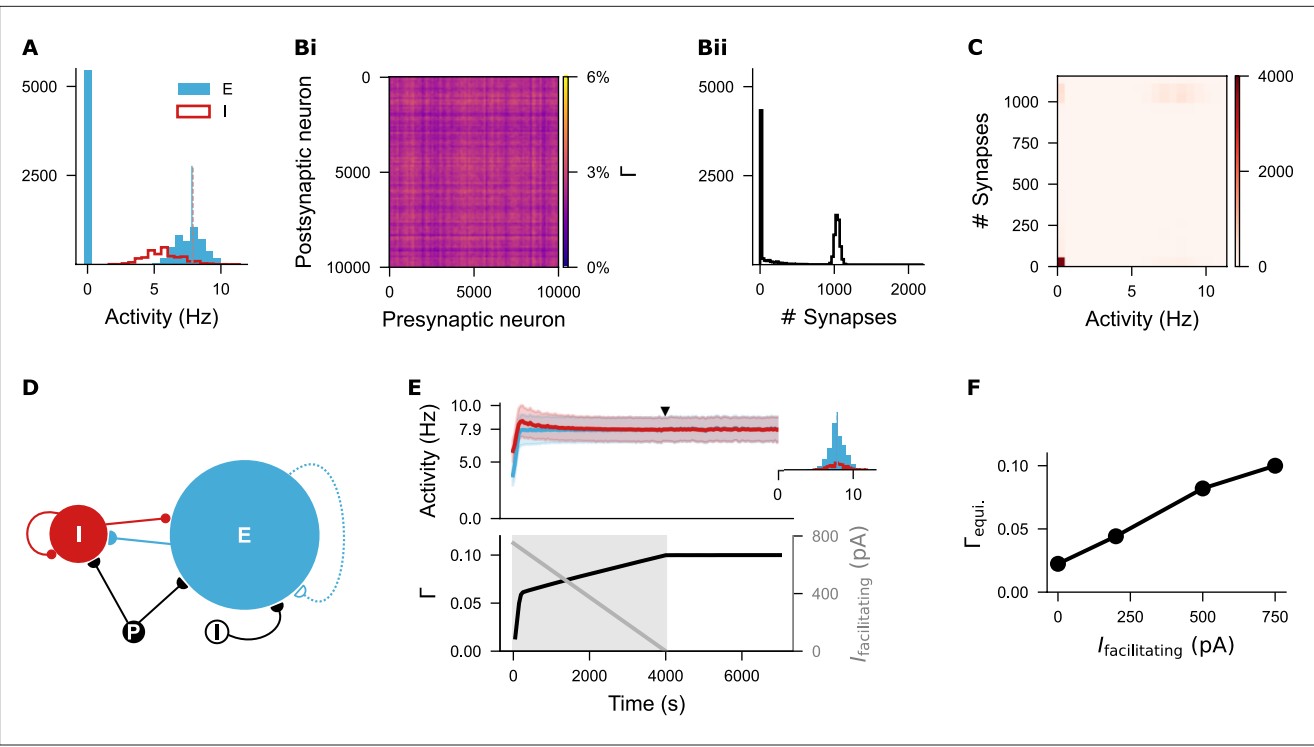

**Figure 4.** Silent neurons remain isolated under the biphasic Gaussian rule. (**A**) Histogram of firing rates for excitatory and inhibitory neurons sampled at the time point indicated in *Figure 3Diii*. Nearly half of the excitatory neurons remained silent. The blue vertical line marks the mean firing rate of non-silent excitatory neurons, while the orange dashed line indicates the target firing rate ($\epsilon = 7.9\,\text{Hz}$). (**Bi-Bii**) Network connectivity matrix and distribution of synapse numbers across individual excitatory neurons. (**C**) Correlation heatmap showing the relationship between neural activity and synapse number per excitatory neuron. Silent neurons failed to form synapses, whereas active neurons fired around the target rate and formed approximately 1000 synapses with other active excitatory neurons. (**D**) Network architecture after introducing a damping facilitating current ($I_{\text{facilitating}}$) to promote network development. (**E**) Temporal dynamics of firing rate (mean and standard deviation) and network connectivity after current injection. The inset displays firing rate distributions for excitatory and inhibitory neurons at the time point indicated by the solid triangle. The facilitating current started at 750 pA and decayed linearly to zero over 4000 s. (**F**) Different initial amplitudes of the facilitating current led to distinct levels of network connectivities. An initial value of 750 pA was used consistently throughout the study.

*2022*), the linear-rule-guided network developed smoothly into a homogeneous, sparsely connected network (10% of connection probability, equivalent to approximately 1000 excitatory synapses per neuron) with an average firing rate near the target value of $\epsilon = 7.9\,\text{Hz}$ (*Figure 3Di*). The Gaussian rule with a zero setpoint ($\eta = 0$) also guided the network to a similar equilibrium state (*Figure 3Dii*). However, the Gaussian rule with two non-zero setpoints ($\eta = 0.7$ and $\epsilon = 7.9$) initially failed to stabilize network development, resulting in network fragmentation and activity silencing (*Figure 3Diii*).

Closer examination of the neural firing rates and network connectivity at the late growth stage of this network (*Figure 3Diii*) revealed that half of the excitatory neurons were silent (*Figure 4A*), and the network exhibited inhomogeneous connectivity, with nearly half of the neurons isolated (*Figure 4Bi–Bii*). Correlating the firing rates with synapse numbers of individual excitatory neurons confirmed that the silent neurons were isolated, while neurons with firing rates close to the target possessed approximately 1000 synapses, similar to the linear rule (*Figure 4C*). These results align with the properties of the Gaussian growth rules: one setpoint is stable ($\epsilon = 7.9$), while the second setpoint can be zero if the synapse numbers remain unchanged after silencing ($\eta = 0$) or non-zero if disconnection occurs after silencing ($\eta > 0$). We refer to the Gaussian rule with one zero-valued setpoint as the *zero Gaussian*

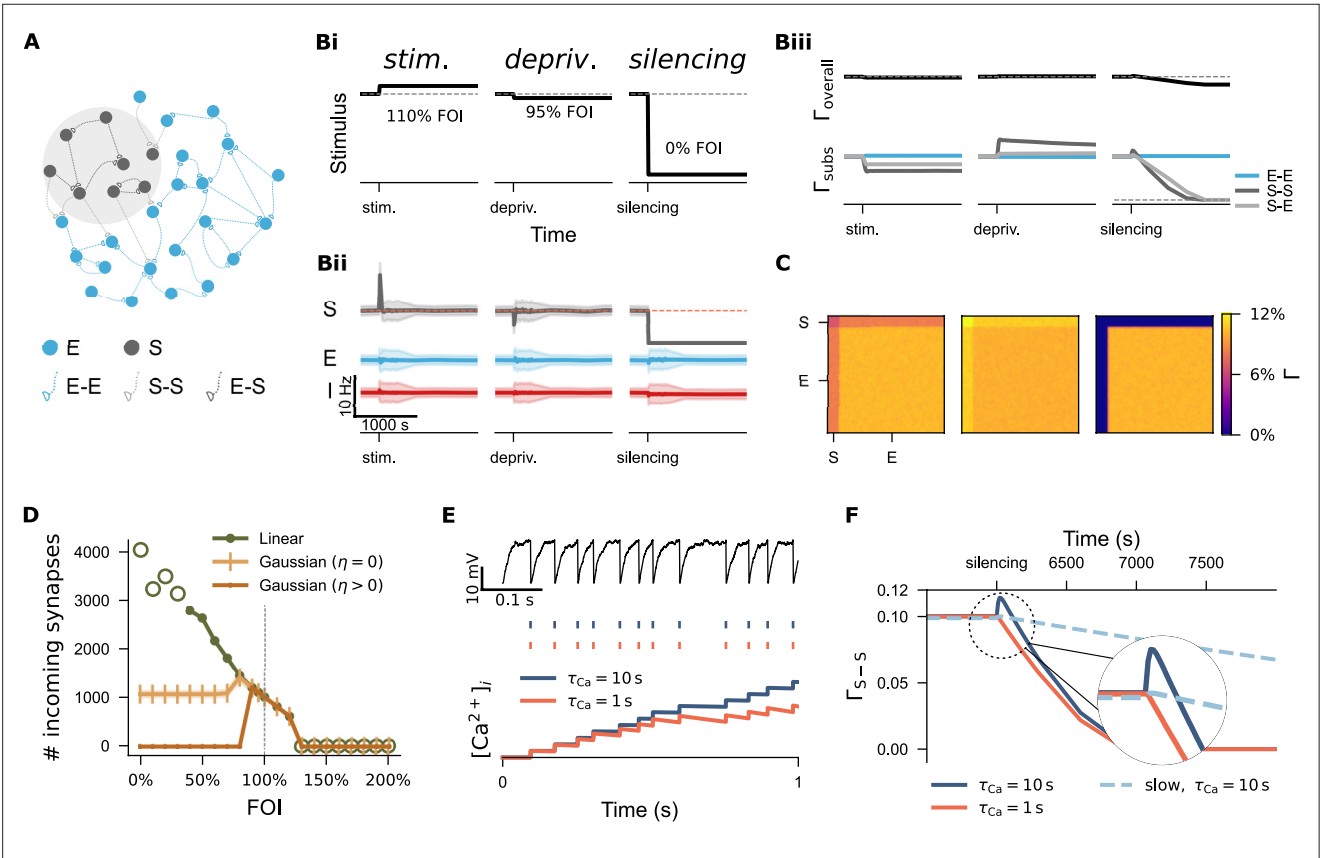

**Figure 5.** Divergent regulation of network connectivity under stimulation and deprivation via three structural plasticity rules. (**A**) A subpopulation (S) comprising 10% excitatory neurons (E) was subject to activity perturbation. All E-E, S-S, and E-S synapses were governed by the biphasic Gaussian rule. (**Bi**) Activity perturbation protocol. Three levels of input intensity, expressed as a percentage of the original Poisson input (fold of intensity, FOI), were used: *stimulation* (110% FOI), *mild deprivation* (95% FOI), and *silencing* (0% FOI). (**Bii**) Temporal dynamics of mean firing rates and standard deviation for the S, E, and inhibitory population (I) under each protocol. (**Biii**) Temporal evolution of overall network connectivity ($\Gamma_{\text{overall}}$) and subpopulation-specific connectivity ($\Gamma_{\text{subs}}$) under each condition. Due to the symmetric application of plasticity rules to spines and boutons, the connection probabilities from E to S and S to E are equivalent. Therefore, only S-E traces are shown unless otherwise noted. (**C**) Final network connectivity matrices following stimulation, mild deprivation, or silencing. (**D**) Average number of incoming synapses to neurons in S subpopulation across a range of FOIs. Empty green circles indicate values from networks exposed to extreme stimulation or inhibition, where both activity and network connectivity were unstable. (**E**) Examples of two neurons receiving identical external input but with different calcium decay time constants ($\tau_{\text{Ca}}$). The upper panel shows membrane potential traces; the middle panel displays spike trains, and the lower panel illustrates integrated calcium concentration over time. (**F**) Connectivity trajectories of silenced subnetwork under three different conditions. The dashed circle highlights a region shown at higher magnification.

*rule*, and the variant with two non-zero setpoints is termed the *biphasic Gaussian rule*. After applying appropriate damping current, the biphasic Gaussian rule produced network statistics comparable to those observed with the linear and zero Gaussian rules (*Figure 4D, E and F*), thereby allowing a systematic comparison of their effects following network development.

## The biphasic Gaussian rule reconciles homeostatic and non-homeostatic structural changes following activity perturbation

To examine the first hypothesis—that synapse-number-based structural plasticity responds non-monotonically to activity perturbations—we systematically modulated the input strength to a subgroup of excitatory neurons (10% of the entire excitatory population) after network development. The input strength varied from 0% to 200% fold of the original intensity (FOI, *Figure 5A*), mimicking both input deprivation and stimulation. *Figure 5Bi–Biii and C* illustrates three example protocols demonstrating neural activity and network connectivity under the biphasic Gaussian rule. As expected, stimulation increased the activity of the targeted subpopulation (S) (left panel in Bii) and triggered a reduction in connectivity both among the stimulated neurons (S-S) and between the stimulated and non-stimulated excitatory neurons (S-E, left panel in Biii). The connectivity matrices in panel C display the final network connectivity. Ultimately, homeostatic disconnection restored the activity levels of the stimulated neurons to the setpoint value (orange line in Bii). In contrast, the outcome of activity deprivation depended on its severity. Both mild deprivation and silencing reduced the neural activity of the affected subpopulation (middle and right panels in Bii). However, only mild deprivation allowed the network to restore activity to the setpoint through a homeostatic increase in network connectivity (middle panels in Biii and C). In the case of complete silencing, the affected neurons disconnected from the network (right panels in Biii and C).

A systematic analysis of incoming synapse numbers confirmed the biphasic dependency of the biphasic Gaussian rule (dark yellow curve in *Figure 5D*), in contrast to the monotonic behavior of the linear rule (green curve). The zero Gaussian rule, which features a zero-valued setpoint ($\eta = 0$), exhibited an intermediate profile: external stimulation and mild deprivation triggered homeostatic reductions or increases in synapse numbers, respectively, but strong deprivation left the network silent yet structurally intact (light yellow curve). Among the three rules, the biphasic Gaussian rule best captured the homeostatic characteristics of the linear rule while also allowing for non-homeostatic, silencing-induced spine loss, consistent with our experimental findings and previous studies (*Keck et al., 2008*; *Vuksic et al., 2011*; *Vlachos et al., 2012*; *Vlachos et al., 2013*; *Bissen et al., 2021*).

Lesion- or denervation-induced plasticity is of particular interest in the context of activity-dependent structural plasticity and holds clinical relevance (*Desmurget et al., 2007*; *Di Pino et al., 2014*; *Sampaio-Baptista et al., 2018*). The biphasic Gaussian rule provides a useful model for studying denervation-induced degeneration and subsequent regeneration. However, we observed a small, biologically unrealistic increase in connectivity immediately following silencing, possibly caused by a high growth rate or lingering effects of prior activity on intracellular calcium concentrations. In *Figure 5E*, we show that two neurons receiving identical external input displayed identical membrane potential dynamics and spike trains, but a difference in the calcium time constant ($\tau_{Ca}$) led to a divergence in accumulated calcium concentration. Since synapse-number dynamics in our model depend on intracellular calcium concentration—used here as a proxy for firing rate—a shorter time constant ($\tau_{Ca} = 1\,\text{s}$, orange curve in *Figure 5F*) carried less activity history, resulting in a smoother reduction in network connectivity. A similar effect was achieved by reducing the structural plasticity growth rate to 10% of the original value (light blue dashed curve, *Figure 5F*). Our computer simulations demonstrated that the time scales of calcium dynamics and structural plasticity interact. The decay time constant of intracellular calcium concentration is difficult to estimate, given the diversity of calcium signals and their temporal and spatial compartmentalization. Moreover, structural plasticity typically operates over longer timescales—minutes to days. Therefore, we used a slow growth rate and the original calcium time constant in all subsequent simulations. The faster growth rate employed in *Figure 6D–F* was used solely to accelerate spine turnover and reduce simulation time to simulate long-term effects.

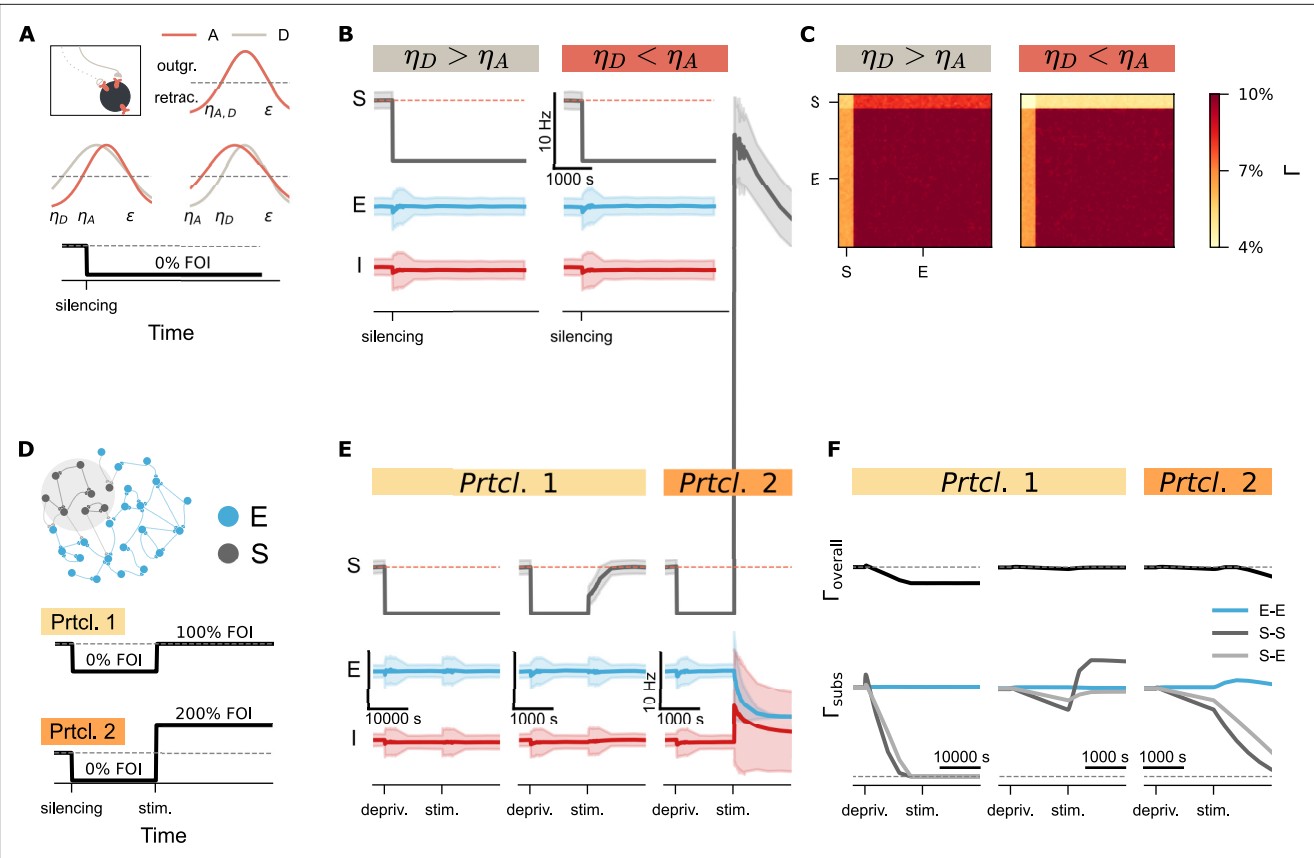

**Figure 6.** Activity perturbation and recurrent connectivity jointly shape the evolution of network connectivity. (**A**) Structural plasticity rules applied to both axonal boutons (A, light brown curve) and dendritic spines (D, pink curve), using identical growth functions with $\eta_A = \eta_D$ (upper inset). Variations with asymmetric parameters (different $\eta$ values could be used for axonal and dendritic elements, such as $\eta_A > \eta_D$ or $\eta_A < \eta_D$) are conceptually possible. A silencing protocol was implemented using a fold of original input intensity (FOI). (**B**) Temporal dynamics of neural activity (mean and standard deviation) for the silenced subpopulation (S), excitatory neurons (E), and inhibitory neurons (I) under two different conditions. (**C**) Final network connectivity matrices at the end of the silencing protocol under the same two conditions shown in **B**. (**D**) Experimental protocols used to assess the combined effects of recurrent connectivity and external stimulation. External stimulation was introduced to the deprived subnetwork under conditions of varying recurrent connectivity. In Protocol 2 (Prtcl. 2), the external stimulation intensity was doubled relative to Protocol 1 (Prtcl. 1). The same structural plasticity rule was applied to both axonal and dendritic elements ($\eta_A = \eta_D$). (**E**) Temporal trajectories of neural activity (mean and standard deviation) following silencing and external stimulation. The network in the left panel was simulated with a structural growth rate 10 times faster than that in the middle and right panel, resulting in lower recurrent connectivity. (**F**) Corresponding evolution of overall connectivity ($\Gamma_{\text{overall}}$, top) and subnetwork-specific connectivity ($\Gamma_{\text{subs}}$, bottom) for excitatory-excitatory (E–E), subpopulation (S–S), and cross-population (S–E) connections.

## Recurrent connectivity and activity perturbation shape synapse-number dynamics during structural recovery

While silencing-induced disconnection captures the characteristic spine loss observed during input deprivation, complete neural isolation is rarely observed *in vivo*. Instead, spine recovery often follows an initial loss, consistent with experimental observations and previous modeling studies (*Keck et al., 2008*; *Vlachos et al., 2012*; *Bissen et al., 2021*). Prior simulations using Gaussian rules have also reported such 'physiological' recovery, assuming distinct growth rules for axonal boutons and dendritic spines (*Butz and van Ooyen, 2013*; *Butz et al., 2014*). However, when we implemented different $\eta$ values for axonal bouton and dendritic spine growth in a topology-free network (without distance-dependent connectivity), synapse numbers did not recover after silencing (*Figure 6A and B*). Instead, we observed only asymmetric connectivity between input and output synapses of the deprived subpopulation (*Figure 6C*). Closer examination of the network revealed the critical role of recurrent input. In networks with distance-dependent connectivity, neurons at the edge of the deprived region remained partially active due to input from non-deprived neighbors, whereas neurons at the center became fully isolated—even though external input was uniformly removed.

This observation prompted further investigation into the role of recurrent connectivity in synaptic recovery. In our topology-free network, we systematically modulated both external input and internal connectivity. After applying the silencing protocol, we introduced external stimulation to the deprived subnetwork under conditions of varying recurrent connectivity. In *Figure 6E and F* (Protocol 1), the network shown in the left panel was simulated with a ten-fold faster timescale than the middle panel, resulting in lower recurrent connectivity at the time stimulation was applied. To isolate the effect of recurrent connectivity, we avoided varying stimulation parameters—such as intensity, duration, interval, or session number—which have been addressed in prior work (*Lu et al., 2019*; *Manos et al., 2021*; *Anil et al., 2023*). Instead, we kept stimulation constant and compared two intensities as a proof of concept. The results demonstrated that network recovery depends on the product of external input and recurrent connectivity. In the left panel of *Figure 6E*, the deprived subnetwork remained silent due to low recurrent connectivity, despite receiving stimulation. In contrast, higher recurrent connectivity in the middle panel allowed the same stimulation to reactivate the subnetwork. Connectivity traces in *Figure 6F* confirmed that residual recurrent input amplified the stimulation, enabling synapse regeneration and rewiring. However, when the external stimulation was too strong (protocol 2), the amplification effect of recurrent connectivity pushed the system into a different regime: from silencing-induced degeneration to over-excitation-induced homeostatic degeneration (*Figure 6E and F*, right panels). These findings suggested that recurrent connectivity serves as a critical amplifier of external input, shaping structural recovery or degeneration following activity perturbations.

## Synaptic scaling modulates structural plasticity and reshapes connectivity during activity perturbation

So far, we have confirmed that the biphasic Gaussian rule reconciled both homeostatic and non-homeostatic regulation of spine numbers under activity deprivation, as proposed in the first hypothesis, and that recurrent connectivity plays a critical role in shaping the expression of this rule. In line with this framework, homeostatic synaptic scaling may dynamically modulate neural activity and reshape recurrent connectivity by adjusting functional synaptic transmission—forming the basis of the second hypothesis. Specifically, downscaling of excitatory synapses may reduce heightened neural activity, potentially masking structural changes if it acts rapidly and effectively. Conversely, homeostatic strengthening of excitatory synapses under silencing may restore activity, thereby shifting structural plasticity from non-homeostatic synapse loss to homeostatic rewiring—similar to the effects observed with external stimulation or distance-dependent connectivity.

To test this hypothesis, we implemented a monotonic, activity-dependent synaptic scaling rule, guided by the same intracellular calcium concentration as the biphasic Gaussian structural plasticity rule. To ensure consistent baseline connectivity and avoid developmental confounds, we first allowed the network to grow using the biphasic Gaussian structural rule until it reached equilibrium. Synaptic scaling was then activated at the onset of the activity perturbation protocol (*Figure 7A*, *Figure 7—figure supplement 1A*). To disentangle the contributions of structural plasticity and synaptic scaling, we analyzed two connectivity measures: (1) *Structural Connectivity* ($\Gamma_{struc.}$), determined solely by synapse number, and (2) *Effective Connectivity* ($\Gamma_{effec.}$), defined as the product of synapse numbers and synaptic weights. Without synaptic scaling, synaptic weights remained uniform, so structural and effective connectivity were identical (*Figure 7C*, left panels; *Figure 7—figure supplement 1Bi*). Intuitively, when neural activity increased through stimulation, both structural plasticity and synaptic scaling acted redundantly in a homeostatic manner. In this case, synaptic downscaling attenuated homeostatic disconnection, making spine loss less pronounced (*Figure 7—figure supplement 1Bii*).

In contrast, activity deprivation revealed a more complex interaction. With weak synaptic scaling (*Figure 7B and C*, $\rho = 0.01$), effective connectivity to the deprived neurons (E-S) increased gradually but had minimal impact on firing rates and failed to restore synapse numbers and network connectivity. When scaling strength was increased ($\rho = 0.02$; *Figure 7C*, right panel), effective E-S connectivity increased more substantially, reactivating previously silent neurons and initiating synapse regeneration and rewiring. We further examined connectivity matrices at two time points: $t2$ (*Figure 7Di, ii*) and $t3$ (*Figure 7Ei, ii*). Notably, at $t2$, input synapse numbers to the deprived subgroup decreased, but the remaining synapses receiving input from non-deprived neurons increased in synaptic weight. This led to the reactivation of silent neurons (*Figure 7B*, right panels), and by $t3$, input synapse numbers began to recover. These results confirm that monotonic synaptic scaling can oppose the biphasic

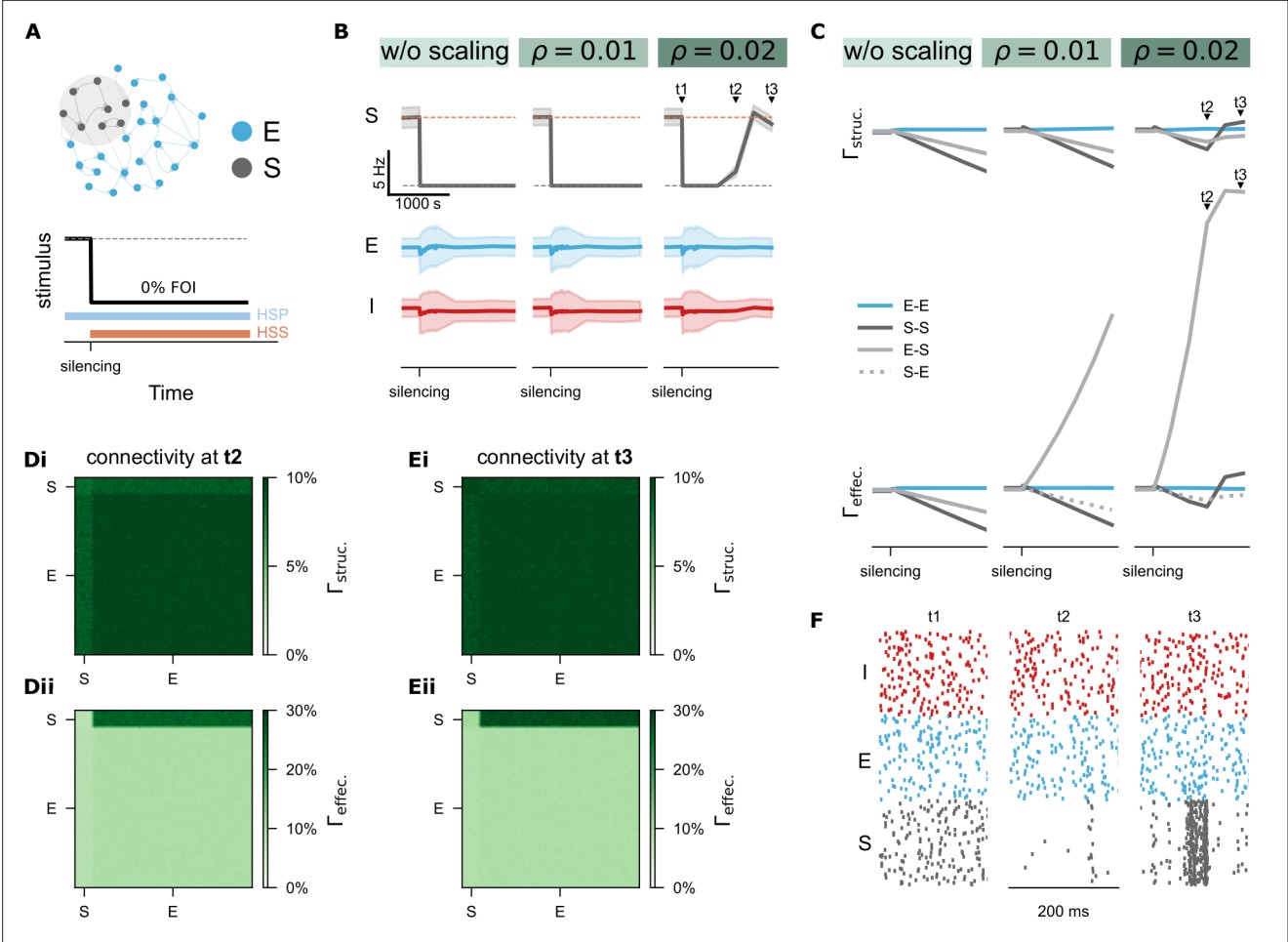

**Figure 7.** Homeostatic synaptic scaling (HSS) shapes effective connectivity and interacts with homeostatic structural plasticity (HSP). (**A**) Protocol of silencing (0% FOI) was applied to a subpopulation of excitatory neurons (S) and synaptic scaling was activated with three different strengths: $\rho = 0$ (no scaling), $\rho = 0.01$ (weak scaling), and $\rho = 0.02$ (strong scaling). (**B-C**) Time courses of network activity (mean and standard deviation) and network connectivity. $\Gamma_{\text{struc.}}$ denotes structural connectivity and $\Gamma_{\text{effec.}}$ refers to effective connectivity. (**Di-Dii**) Structural and effective connectivity matrices of the entire network at time point $t2$. (**Ei-Eii**) Structural and effective connectivity matrices at time point $t3$. (**F**) Raster plots of spiking activity for 100 representative neurons at the time points $t1$, $t2$, and $t3$, including inhibitory (I, red), excitatory (E, blue), and silenced excitatory neurons (S, grey).

The online version of this article includes the following figure supplement(s) for figure 7:

**Figure supplement 1.** Time courses of network activity (Bi), structural connectivity (Bii, upper panels), and effective connectivity (Bii, lower panels) under the stimulation protocol, with and without the homeostatic synaptic scaling rule.

structural plasticity rule under strong deprivation, facilitating a transition from synapse loss to activity-dependent rewiring. Intriguingly, with sufficient scaling strength, both synapse numbers and average neural activity were restored after silencing, and the network's temporal dynamics were altered. The previously deprived subpopulation showed high synchronization after recovery (*Figure 7F*), suggesting mechanisms relevant to pathological states such as post-traumatic epilepsy. Overall, our simulations suggested that biphasic HSP and monotonic homeostatic synaptic scaling play complementary and partially redundant roles in maintaining firing rate homeostasis. Their interaction provides a biologically plausible mechanism for reconciling stable activity control with flexible network reorganization.

## Spine size dynamics reveal an interaction between structural plasticity and synaptic scaling

The altered dynamics in the deprived neural population after recovery may result from dominant input originating from non-deprived excitatory neurons. To test this, we analyzed the firing rates and input synaptic weights of representative neurons from both the deprived (grey) and non-deprived

(blue) excitatory populations (*Figure 8A*). This analysis revealed a marked increase in synaptic weights from non-deprived to deprived neurons (*Figure 8A*, lower panel), further supported by a global shift in synaptic weight distribution (*Figure 8B*). Additionally, the distribution analysis indicated changes in input synapse numbers after silencing (see values above *Figure 8B*), prompting us to assess both synaptic weights and input synapse numbers for a representative neuron (solid line) and its peers (dashed line in *Figure 8C*) to gain further insight into structural plasticity. Consistent with experimental findings, input synapse numbers of deprived neurons decreased after silencing but gradually recovered. Interestingly, recovery patterns diverged: synapses from reactivated neurons exceeded their initial numbers (upper panel), while those from non-deprived neurons (middle panel) and total synapse counts (lower panel) returned to slightly lower levels than baseline. These results demonstrated that an appropriate combination of homeostatic synaptic scaling and structural plasticity can replicate the process of spine loss and recovery following activity deprivation. Importantly, the model captures a commonly observed outcome of chronic activity silencing: fewer but stronger synapses, corresponding to fewer but larger spines.

To test whether this principle also holds in our NBQX experiments—and to assess potential dose dependency—we analyzed spine sizes before and after chronic NBQX treatment in mouse entorhinal-hippocampal slice cultures. As expected, initial spine size distributions in all three groups showed a long-tail pattern, with a predominance of small spines (*Figure 8Di–Diii*), consistent with previous reports (*Arellano et al., 2007*; *Buzsáki and Mizuseki, 2014*). After 3 days of treatment, the distribution of spine sizes changes ($\Delta$ spine size) approximated a normal distribution (*Figure 8—figure supplement 1Bi, iii*). At the population level (*Figure 8E*), significant increases in raw spine sizes were observed following 200 nM NBQX treatment ($p < 0.001$, Wilcoxon test; 99.9% CI = [9.09, 25.8], LMM), while 50 µM NBQX treatment led to significant reductions ($p < 0.001$, Wilcoxon test; 99.9% CI = [9.09, 25.8], LMM), indicating a non-monotonic, activity-dependent response. When size changes were normalized by their initial spine sizes, a general increase was observed across all three groups (*Figure 8E* inset), likely driven by the growth of many small spines. Sorting spine size changes by initial size (*Figure 8F*) revealed that in control segments (black curve), small spines generally enlarged while large spines tended to shrink. Under partial inhibition with 200 nM NBQX (orange curve), this relationship was shifted upward, promoting spine enlargement regardless of initial size. However, complete inhibition with 50 µM NBQX (light blue curve), not only reversed the typical growth of small spines but also altered the fate of a subset of large spines—favoring further enlargement instead of the typical shrinkage observed in medium-sized spines.

These findings suggested a homeostatic adjustment of spine sizes in response to partial and complete inhibition. Notably, under complete inhibition, the spine size changes were strongly dependent on initial spine sizes, paralleling both experimental and simulation results in which complete activity blockade resulted in fewer but enlarged spines. However, this pattern deviates from the expected synaptic weight changes predicted by the synaptic scaling rule. This discrepancy will be further examined in the discussion, where we integrate electrophysiological and imaging data for a more comprehensive interpretation.

## Hybrid combinations of synaptic scaling and structural plasticity reconcile spine density inconsistencies

Having obtained both spine density and size data for individual segments, we explored whether a unifying principle might link changes in spine size and density. We quantified the average spine size change ($\bar{\Delta}$ spine size) for each segment and correlated these values with baseline spine densities (*Figure 9A*) and with changes in spine density ($\Delta$ spine density, *Figure 9B*). A significant positive correlation between spine size changes and baseline spine densities was observed in the 200nM NBQX treatment group (*Figure 9A*, $r(24) = 0.65$, $p = 0.00064$, Pearson's correlation; *Figure 9—figure supplement 1A*), but not in the control or 50 µM NBQX groups. When we correlated spine size changes with spine density changes across all segments, weak inhibition (200 nM NBQX) typically resulted in coordinated increases in both size and density (see orange dots clustered in the upper right quadrant in *Figure 9B*). This suggested that spine-dense segments may be more responsive to weak inhibition, reflecting a synergistic interaction between structural and functional plasticity. In contrast, under strong inhibition (with 50 µM NBQX), the size-density relationships were highly variable (light blue dots in *Figure 9B*), consistent with the diverse spine remodeling outcomes observed.

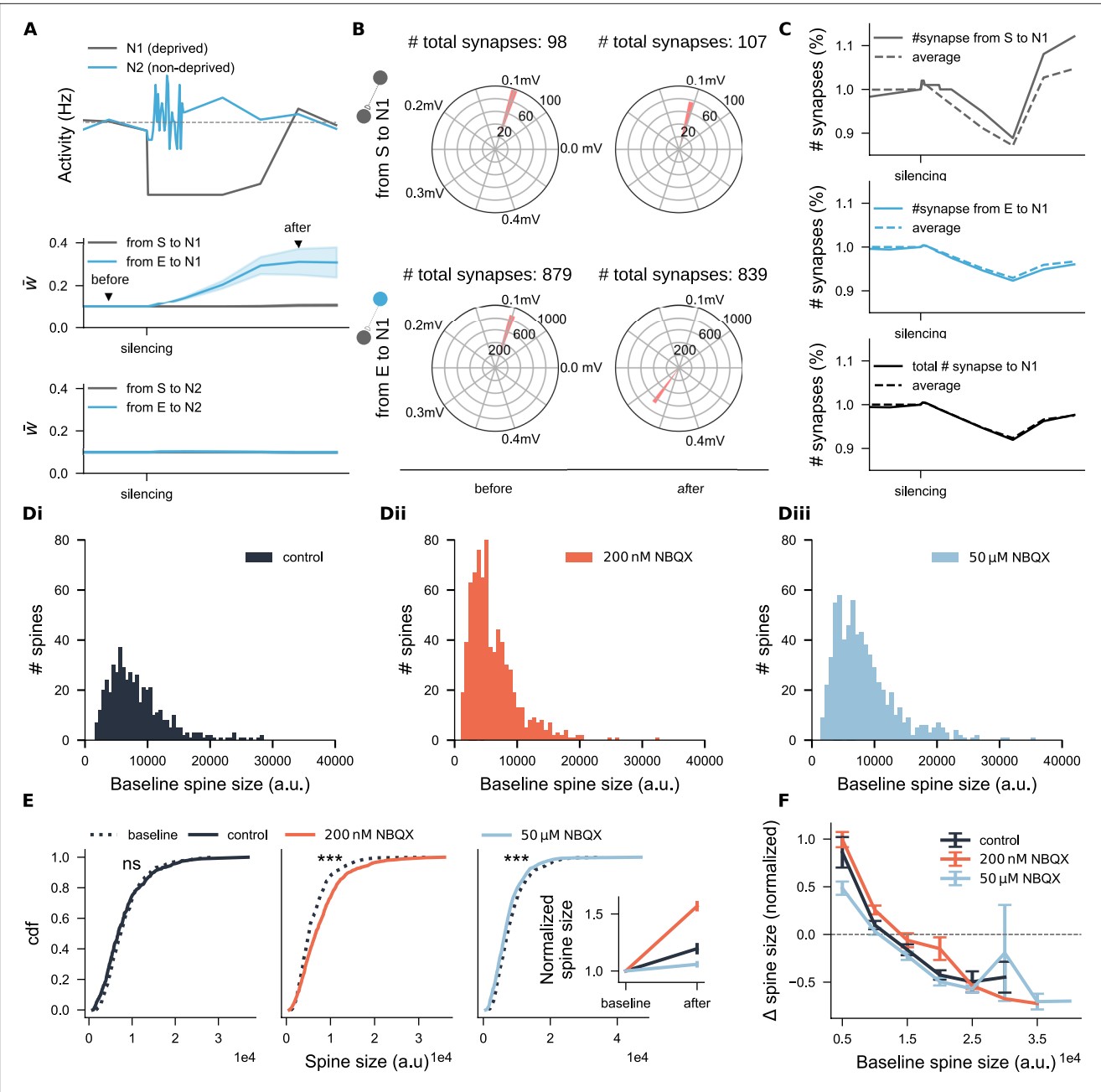

**Figure 8.** Spine size analysis suggests interactions between functional and structural plasticity. (**A**) Activity trajectories and average incoming excitatory synaptic weights of two representative excitatory neurons: one from the deprived subpopulation (N1, grey) and one from the non-deprived group (N2, blue). The gray dashed line in the upper panel indicates the target activity setpoint. Shaded areas represent the standard deviation of incoming synaptic weights. Synapses were categorized based on their source: from the deprived subpopulation (S, grey) or from non-deprived excitatory neurons (E, blue). Triangles in the middle panel indicate the time points used for synaptic weight distribution analyzed in **B**. (**B**) Distributions of synaptic weights for neuron N1 *before* and *after* activity deprivation. Numbers above the plots indicate the total synapse count per source type at each time point. (**C**) Normalized synapse numbers over time of neuron N1 (solid line) and the averaged across all deprived neurons (dashed line). (Di-Diii) Baseline spine size distributions for three groups: control (N=489), 200 nM-NBQX treated (N=736), and 50 μM-NBQX treated (N=675). (**E**) Cumulative distribution functions (cdf) of spine sizes before and after the three-day treatment. The inset shows average normalized spine sizes across conditions. (**F**) Normalized spine size changes grouped by initial spine size. The $x$-axis denotes the upper limit of each bin. Data points above the dashed line represent spine enlargement; points below indicate shrinkage. Error bars in the inset of **E** and in **F** represent SEM for each group.

The online version of this article includes the following figure supplement(s) for figure 8:

**Figure supplement 1.** Distributions of baseline spine sizes and spine size changes over the three-day treatment period across all three groups.

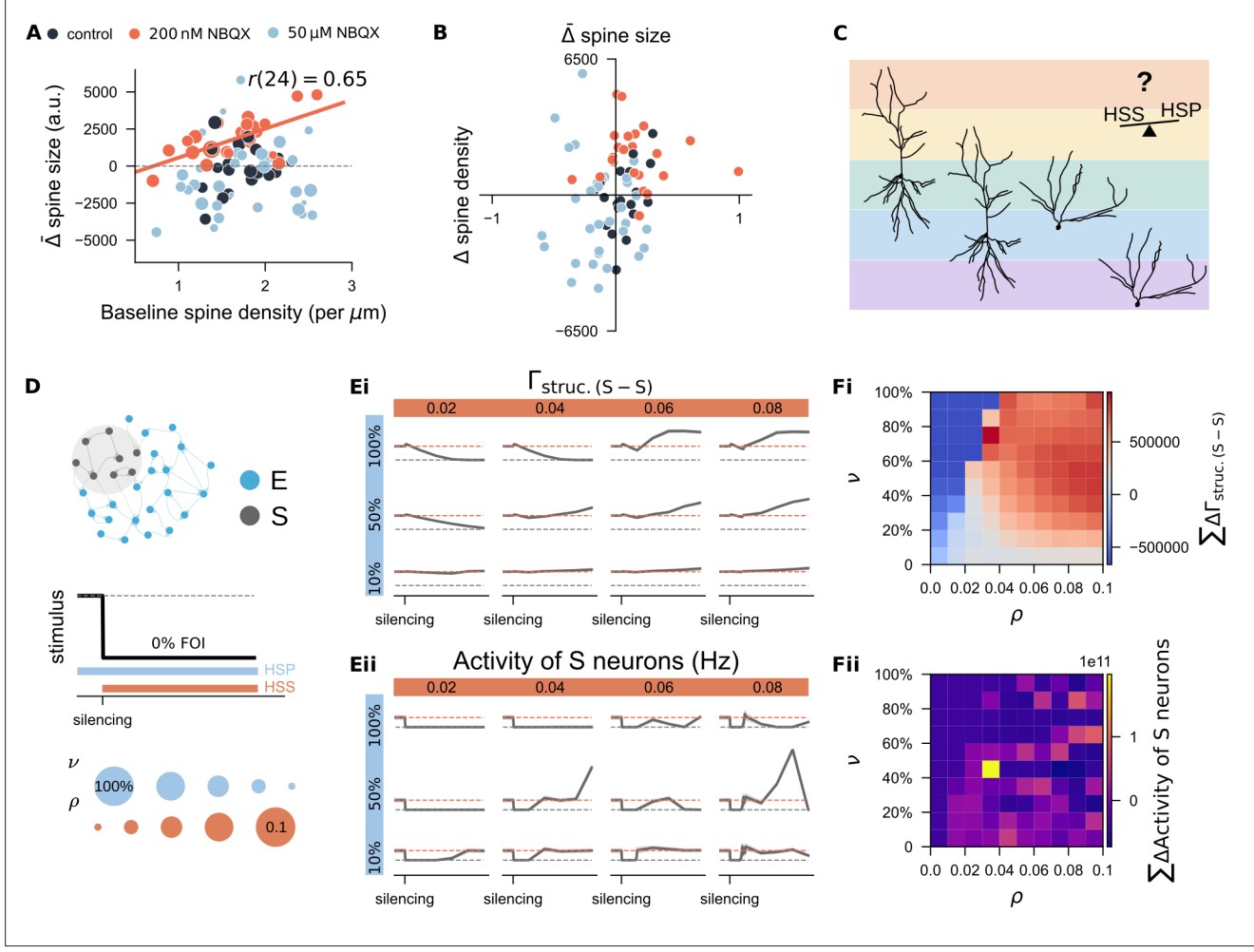

**Figure 9.** Systematic study of the interaction between homeostatic synaptic scaling and structural plasticity in response to activity silencing. (**A**) Correlation between each dendritic segment's average change in spine size and its initial spine density. Marker size reflects the net change in spine density over the three-day period. (**B**) Correlation between each dendritic segment's average changes in spine size and its change in spine density. (**C**) Conceptual hypothesis: Different combinations of homeostatic synaptic scaling (HSS) and homeostatic structural plasticity (HSP) may form a spectrum, allowing individual neuron types or even distinct dendritic segments within the same neuron to be governed by unique rule combinations. This framework could account for the variability observed in experimental studies of structural plasticity. Sample neuron reconstructions are based on CA1 pyramidal cells (left two) and dentate gyrus granule cells (right two) recorded in our lab. (**D**) Simulation protocol used to explore the interaction between homeostatic synaptic scaling and structural plasticity. The growth rate of the HSP rule ($\nu$) and the strength of the HSS rule ($\rho$) were systematically varied. (**Ei-Eii**) Example time course of structural connection probability and neural activity in the deprived subpopulation (S) under selected combinations of ($\nu$) and ($\rho$). (**Fi-Fii**) Quantification of deviations in connectivity and firing rate under different parameter combinations. Deviations were computed as the area between the actual trajectory and either the equilibrium connection probability (10%) or the target firing rate ($\epsilon$) over the silencing period. Results represent the mean from 11 independent trials.

The online version of this article includes the following figure supplement(s) for figure 9:

**Figure supplement 1.** Relationship between average spine size change and initial spine density for individual dendritic segments.

**Figure supplement 2.** Schematic representation of the plasticity rules used in this study, framed within a control theory perspective.

**Figure supplement 3.** Quantification of $\Delta$Activity in S neurons and $\Delta\Gamma_{\text{struc.(S-S)}}$ used to generate the heatmap shown in *Figure 8E and F*.

Although our imaging data did not reveal a universal rule, they suggested that responses to activity deprivation are highly segment-specific. This led us to hypothesize that different combinations of homeostatic synaptic scaling and structural plasticity operate at the dendritic segment level, or may even vary between cell types (*Figure 9C*). Such diversity could help explain inconsistent findings across brain regions or between apical and basal dendrites of the same neuron (*Lenz et al., 2023b*).

To test this hypothesis, we simulated the same silencing protocol while varying the structural plasticity growth rate ($\nu$) and synaptic scaling strength ($\rho$) (*Figure 9D*). Example trajectories of neural activity and structural connectivity in the deprived subpopulation under different conditions are shown in *Figure 9Ei, ii*. To quantify rewiring efficacy, we calculated the cumulative deviation of structural connectivity from the equilibrium value (10%) in the deprived subnetwork over the silencing period (*Figure 9Fi*). Cold colors indicate failed connectivity restoration; warm colors indicate successful or excessive rewiring. Observationally, slower growth rates preserved more residual synapses, allowing even weak synaptic scaling to restore connectivity. In contrast, faster growth resulted in more extensive spine loss, requiring stronger synaptic scaling for recovery.

We also tracked deviations of firing rates from target (7.9 Hz) over the same period (*Figure 9Fii*). Parameter combinations that failed to restore network connectivity (*Figure 7Fi*) showed persistent silencing (dark blue in *Figure 7F*). However, successful synapse regeneration (warm colors in *Figure 7Fi*) did not always ensure optimal firing rate recovery. Best results were achieved with moderate synapse regeneration (light pinkish colors in panel Fii), whereas strong structural growth combined with strong synaptic scaling could induce excessive activity (bright colors) or delayed rebound activity (dark blue colors). In conclusion, our systematic simulations demonstrated that calcium-based homeostatic synaptic scaling and structural plasticity can interact in a dynamic, context-dependent manner. Their interplay—shaped by timing and strength—offers a plausible explanation for the diverse spine remodeling outcomes observed experimentally (*Moulin et al., 2022*).

## Discussion

The mammalian brain is a complex system composed of billions of neurons and non-neuron cells, organized into distinct but interconnected layers, regions, and circuits (*Herculano-Houzel, 2012*). It is remarkably robust—information overload does not result in catastrophic forgetting (*French, 1999*), nor does sensory deprivation disrupt overall network dynamics (*Hengen et al., 2016*; *Ma et al., 2019*; *Torrado Pacheco et al., 2019*). Hebbian plasticity and homeostatic synaptic scaling (HSS), two well-characterized forms of functional plasticity, are central to preserving this stability. In contrast, structural plasticity has received comparatively less attention, owing to inconclusive experimental findings (*Moulin et al., 2022*) and a lack of comprehensive computational models (*Butz et al., 2009b*). Under conditions of activity deprivation, both homeostatic and non-homeostatic changes in spine numbers have been reported (*Moulin et al., 2022*), challenging the explanatory power of the HSP framework. In this study, we confirmed a biphasic HSP rule with experiments and examined its interaction with a monotonic HSS mechanism during activity deprivation using computer simulations. Our time-lapse imaging experiments confirmed the non-monotonic, activity-dependent regulation of spine numbers, supporting the proposed biphasic HSP model. Complementary simulations further demonstrated that this rule reconciles opposing structural outcomes—spine gain and loss—and interacts with HSS to jointly regulate synapse number and synaptic strength. We argue that the redundancy and heterogeneity between HSP and HSS mechanisms are critical for maintaining firing rate homeostasis and ensuring long-term network stability.

Our research significantly advances the understanding of structural plasticity by providing direct evidence of its activity-dependent and differential regulation. While structural plasticity is frequently linked to cortical reorganization following injury, amputation, or disease-related cognitive decline, its theoretical modeling has lagged behind that of functional plasticity—largely due to challenges of capturing high-dimensional morphological dynamics. Among existing mathematical frameworks, some have focused on dendritic tree development (*Vormberg et al., 2017*; *Ferreira Castro et al., 2020*; *Cuntz et al., 2021*), while others emphasized synapse formation and rewiring as key distinctions from functional plasticity models. However, these early models generally adopted a synapse-specific, Hebbian-like approach, relying on use-dependent growth and pruning of individual synapses (*Bourjaily and Miller, 2011*; *Fauth et al., 2015*; *Navlakha et al., 2015*; *Spiess et al., 2016*; *Wang et al., 2021*), which often proved insufficient to stabilize network dynamics. In contrast, Butz and van Ooyen introduced a cell-autonomous HSP rule, grounded in the earlier work of van Ooyen and van Pelt's work (*van Ooyen and van Pelt, 1994*) and conceptually aligned with ideas proposed by *Dammasch, 1989*. This model uses a target firing rate—or intracellular calcium concentration—as a setpoint to regulate spine and bouton formation or elimination (*van Ooyen, 2011*; *Butz and van Ooyen, 2013*). As a synapse-number-based mechanism, it serves as a structural analog to synaptic

scaling for maintaining firing rate homeostasis. At the same time, it retains the associative nature of correlation-based models (*Gallinaro and Rotter, 2018*; *Lu et al., 2019*; *Gallinaro et al., 2022*) and promotes self-organized criticality in developing neural networks (*van Ooyen and Butz-Ostendorf, 2019*). Whether implemented in linear or Gaussian form, this rule has long been proposed as a means of integrating Hebbian and homeostatic plasticity, capturing a range of phenomena observed during development and pathological conditions. Our findings support the biological plausibility of a biphasic variant of this rule, substantiated by both experimental and computational evidence.

We used gradual activity blockade combined with time-lapse imaging to test the hypothesis that spine numbers follow a biphasic relationship with neural activity, thereby addressing longstanding inconsistencies in spine density findings across the literature (*Moulin et al., 2022*). Many experimental paradigms—such as lesion-based models or pharmacological inhibition—introduce confounding factors, including neuroinflammatory cytokine production (*Vlachos et al., 2012*; *Keck et al., 2013*; *Vlachos et al., 2013*; *Barnes et al., 2017*), which can independently influence plasticity mechanisms (*Turrigiano, 2006*; *Becker et al., 2013*; *Kleidonas and Vlachos, 2021*). Non-traumatic deprivation models like dark rearing (*Wallace and Bear, 2004*) or whisker trimming (*Trachtenberg et al., 2002*) result in non-uniform input deprivation across the dendritic arbor, complicating interpretations of spine density changes across compartments. Even precise manipulation, such as pre-synaptic blockade (*Quinn et al., 2019*) or glutamate uncaging near individual spines (*Tong et al., 2021*), may be limited by their homogeneity of effects or lack of circuit specificity. Importantly, even studies using drugs that uniformly target postsynaptic neurons have reported inconsistent effects on spine density (*Bacci et al., 2001*; *Thiagarajan et al., 2005*; *Wierenga et al., 2005*; *Fishbein and Segal, 2007*; *Mitra et al., 2012*; *Hobbiss et al., 2018*), suggesting that the intensity of deprivation may be a key factor. Variables such as drug type (non-competitive vs. competitive), experimental context (*in vivo* vs. *in vitro*), and circuit integrity strongly modulate the degree of inhibition experienced by target neurons. Our previous work in CA1 pyramidal neurons revealed pathway-specific differences in structural, cellular, and molecular responses to synaptic deprivation, highlighting the importance of input origin in shaping plasticity (*Lenz et al., 2023a*). Based on these observations, we hypothesized a biphasic activity-dependent rule: partial activity suppression promotes spine growth, whereas complete inhibition leads to spine loss. Indeed, our time-lapse imaging confirmed this prediction. Spine density increased following partial AMPA receptor blockade (200 nM NBQX) and decreased under complete inhibition (50 µM NBQX). However, we also observed complex changes in spine size under both conditions, suggesting an interaction between functional and structural plasticity, which we discuss in subsequent sections.

The biphasic structural plasticity model, implemented as a Gaussian-shaped growth curve, successfully reconciled the diverse, activity-dependent changes in spine density observed in our experimental data and simulations. With two non-zero setpoints ($\eta > 0$), the model predicted a range of synapse number outcomes—including increases, no changes, or reductions—depending on the degree of activity perturbation. Notably, silencing-induced synapse loss, which cannot be captured by linear (monotonic) models, falls within the predictive scope of the biphasic rule. A key emergent feature of this model is its ability to induce phase transition through modulation of external input or intrinsic connectivity. For example, appropriately timed and scaled stimulation can reverse silencing-induced synapse loss, triggering homeostatic rewiring. This mechanism may have clinical relevance, potentially informing non-invasive brain stimulation strategies aimed at mitigating stroke-induced network disconnection (*Bai et al., 2022*). In parallel, HSS contributes to synaptic recovery by adjusting synaptic weights, consistent with empirical observations of both synapse numbers and synaptic strength. Similar forms of rewiring have been described in previous work by Butz and van Ooyen using the same Gaussian HSP framework (*Butz et al., 2009a*; *Butz and van Ooyen, 2013*; *Butz et al., 2014*). Their studies incorporated distance-dependent connectivity and emphasized the need for distinct growth rules for axonal boutons and dendritic spines. Our ability to replicate such rewiring in a topology-free, homogeneous network supports and extends their findings. It underscores the importance of intrinsic connectivity in governing synapse loss and recovery, even in the absence of predefined spatial constraints. Importantly, our co-simulation of structural and functional plasticity revealed a synergistic interaction between these mechanisms—an aspect often overlooked in computational studies, yet critical for interpreting biological data. Additionally, inhibitory plasticity may further contribute to the physiological regulation of HSP, as suggested by numerical studies

(*Sinha et al., 2015*); however, this remains beyond the scope of the present work and warrants future investigation.

Nevertheless, our model suggests a potential mechanism by which injury-related seizures could emerge following otherwise adaptive recovery. Stroke is a leading cause of seizures and the development of epilepsy in the elderly population (*Beghi et al., 2011*; *Guekht and Bornstein, 2012*; *Galovic et al., 2021*) with higher seizures incidence reported after hemorrhagic compared to ischemic stroke (*Bladin et al., 2000*). According to our model, cell death and resulting input deprivation initiate a cascade of plastic reorganization aimed at restoring network activity. While this reorganization supports functional recovery, it may also create a substrate for epileptiform activity. In turn, seizure activity can exacerbate neural damage, creating a vicious cycle of degeneration and maladaptive plasticity. A closer examination of the epidemiological data reveals a more nuanced relationship between plasticity and post-stroke seizures. Seizures occurring after stroke are typically classified by onset time: acute symptomatic seizures (or early seizures) occur within seven days of infarct and are linked to metabolic or toxic effects of stroke, whereas remote symptomatic seizures (or late seizures) arise after one week and carry a higher risk of developing into chronic epilepsy (*Galovic et al., 2021*). Although structural and functional plasticity occur within minutes to days, the timing of late seizures aligns more closely with prolonged, plasticity-driven network remodeling. Additional risk factors—such as younger age (<65 years) (*Phan et al., 2022*) and cortical involvement (*Galovic et al., 2018*)—have been associated with both increased seizure susceptibility and heightened neural plasticity. These insights suggest that plasticity itself may contribute to post-stroke epileptogenesis. Accordingly, interventions that modulate neural plasticity such as antiepilepsy drugs (*Chang et al., 2022*; *Ryu et al., 2024*) or non-invasive brain stimulation (*Adeyemo et al., 2012*; *Aderinto et al., 2024*) hold promise not only for post-stroke rehabilitation but also for seizure prevention in at-risk populations.

In this context, we examined how the combined action of HSP and HSS shapes the interpretation of experimental results. In our NBQX-treated slice cultures, we observed a simultaneous increase in spine size and density after three days of partial inhibition (200 nM NBQX). This enlargement of spines likely enhanced the stability and detectability of small spines in imaging, suggesting that the apparent increase in spine density reflects a combination of spinogenesis, increased spine stability, and potentiation of previously small (and silent) spines. Similar interpretational challenges arise in studies of synaptic scaling results. For example, *Turrigiano et al., 1998* reported changes in miniature EPSC amplitudes (but not frequencies) following chronic inhibition and excitation, inferring synaptic weight scaling. However, other studies have documented concurrent changes in both amplitude and frequency. One explanation is the unsilencing of synapses, whereby formerly silent synapses become active (*Nakayama et al., 2005*), increasing event frequency but lower mean EPSC amplitude due to their relatively small weights. This process may reflect synapse-number-based structural plasticity, which may not be easily detectable with standard imaging methods, depending on variables such as laser intensity, z-resolution, or exposure time. Additionally, under conditions of complete inhibition (50 μM NBQX), we observed distinct and opposing size changes: small spines tended to shrink or disappear, while large spines further enlarged, resulting in an overall increase in sEPSC amplitudes. Such differential response could explain discrepancies between electrophysiological and imaging data. Together, these findings suggest that spine numbers and synaptic weights represent complementary facets of a broader, integrated concept of 'effective synaptic transmission'. Relying on only one of these measures risks misrepresenting the true functional state of the network.

Our systematic study successfully replicated the empirical observation that chronic activity inhibition leads to spine loss, with the remaining spines often exhibiting enlarged heads (*Thiagarajan et al., 2005*; *Fishbein and Segal, 2007*). This phenomenon can be explained by a shared calcium signaling mechanism underlying both HSP and HSS rules. While HSP consistently uses intracellular calcium concentration as a setpoint, most HSS models instead rely on firing rate setpoint (*Tetzlaff et al., 2011*; *Tetzlaff et al., 2012*; *Tetzlaff et al., 2013*; *Herpich and Tetzlaff, 2019*), synaptic weight normalization (*Sullivan and de Sa, 2006*), or presynaptic mechanisms (*Liu and Buonomano, 2009*; *Liu, 2011*). Notably, one earlier model introduced an integral feedback control algorithm (*van Rossum et al., 2000*), using a variable with an exponential kernel to track neural spiking activity and adjust synaptic weights. At that time, intracellular calcium had not yet been identified as the principal variable tracking neural activity (*Grewe et al., 2010*). Building on this idea, we reinterpreted the model's control variable using a calcium-based variant, allowing us to examine the interaction between the HSP and HSS

more mechanistically. In this way, our study also served as a Gedankenexperiment—a conceptual experiment—to explore how structural and functional plasticity interact. If both rules are redundantly monotonic, the one with the faster timescale will dominate. However, when a biphasic HSP rule is combined with a monotonic HSS rule, our simulations showed divergent behavior under strong inhibition: neurons exhibit reduced spine density and enlarged spine heads, mimicking experimental observations. From an energy efficiency standpoint, synapse formation and spine growth are constrained by protein synthesis and ATP availability. It is thus energetically more favorable to potentiate existing synapses than to form new ones. This trade-off supports the complementary roles of HSP and HSS in maintaining network activity. This interplay is also supported by recent theoretical work on dendritic trafficking (*Aljaberi et al., 2021*) and studies of morphological changes in Alzheimer's disease (*Bachmann et al., 2020*). Our findings highlight the interdependency between HSP and HSS, which may underlie divergent results across brain regions, cell types, dendritic compartments, and experimental settings. In particular, the hybrid combination of structural and functional plasticity across dendritic segments supports the emerging view of dendritic compartmentalization (*Sutton et al., 2006*; *Branco et al., 2008*; *Makara et al., 2009*), a principle that is critical to understanding plasticity but remains underrepresented in computational models. More computational work addressing these multiscale interactions is both warranted and encouraged.

While our study addressed both HSS and HSP, it is important to note that the use of NBQX is not commonly used in experiments focused specifically on synaptic scaling. NBQX, a competitive AMPA receptor antagonist, enables precise modulation of network activity via dose titration (*Wrathall et al., 1994*), offering greater flexibility than tetrodotoxin (TTX), which globally blocks action potentials by inhibiting sodium channels. This property makes NBQX a valuable tool for activity modulation, as demonstrated in our electrophysiological recordings and supported by prior *in vivo* and *in vitro* studies (*Wrathall et al., 1994*; *Follett et al., 2000*). Notably, a previous study (*McKinney et al., 1999*) reported a reduction in spine density after applying 20 µM NBQX to hippocampal slices for seven days, with surviving spines predominantly exhibiting a stubby-like morphology. This aligns with our observation of decreased spine density and reduced spine size following 50 µM NBQX treatment and raises the possibility of investigating the functional role of surviving spines under chronic activity blockade (*Obashi et al., 2021*). However, the specific binding affinity of NBQX for GluA2-containing AMPA receptors merits further attention. These receptors are implicated in both TTX-induced synaptic scaling (*Gainey et al., 2009*) and glutamate-induced spine protrusion in the presence of TTX (*Richards et al., 2005*). This selectivity introduces complexity when interpreting NBQX effects, particularly in the absence of TTX. Although there is no definitive evidence that NBQX alone disrupts AMPA receptors synthesis or membrane insertion, its broader impact on synaptic scaling and structural plasticity remains to be fully elucidated. Further studies are warranted to clarify these mechanisms and refine the interpretation of NBQX-based manipulations.

In summary, the use of intracellular calcium concentration as a shared control variable in both HSP and HSS rules closely resembles an integral feedback controller (*Figure 9—figure supplement 2*), a widely used engineering approach to achieve robust performance despite external perturbations. In both rules, calcium concentration serves as the control signal to drive connectivity kernel modification via negative feedback, which integrates neural firing rates over time and returns a filtered signal responding to its activity. However, the HSP and HSS rules differ in their detailed mechanism. The HSP rule more closely aligns with a Proportional-Integral-Differential (PID) controller, while the HSS rule operates more like a Proportional-Integral (PI) controller. Besides sharing the integral (I) component, the HSP rule's proportional (P) component is linked to the growth rate of synaptic elements (*Diaz Pier et al., 2016*), which modifies the number of synaptic elements created or deleted per time unit based on a magnitude relative to the difference between the expected and actual calcium concentration. The scaling factor ($\rho$) in the HSS rule regulates the proportional effects of synaptic scaling. Uniquely, the HSP rule includes a derivative (D) component, determined by the steepness of the growth curve, which is determined by the steepness of the growth curve and dictates how quickly or slowly the system approaches the desired neural activity. This derivative term plays a crucial role in ensuring a smooth and robust response to perturbations, preventing biologically implausible overshooting and abrupt oscillations. The effect is especially pronounced in the Gaussian implementation, where the distance to the target firing rate accelerates differently depending on the position on the calcium concentration axis. Although the exact shape of the growth rule in biological HSP remains to be

determined, we propose that a biphasic curve with variable slope offers a redundant and heterogeneous structural backup to synaptic scaling, supporting robust firing rate homeostasis across a range of conditions. This framework may also help anticipate interactions with other plasticity mechanisms, particularly those involving sub- or supra-threshold calcium dynamics (*Graupner and Brunel, 2012*; *Luboeinski and Tetzlaff, 2023*).

# Materials and methods

**Key resources table**

| Reagent type (species) or resource | Designation | Source or reference | Identifiers | Additional information |
|---|---|---|---|---|
| Strain, strain background (mice) | C57BL/6 J mouse | Jackson Laboratory | RRID:IMSR_JAX:000664 | |
| Strain, strain background (mice) | Thy1-eGFP mouse | Jackson Laboratory | RRID:IMSR_JAX:007788 | |
| Peptide, recombinant protein | Streptavidin* | Invitrogen | Cat#: S32354 RRID:AB_2315383 | Post hoc labeling (1:1000) |
| Chemical compound, drug | Paraformaldehyde (PFA) | Carl Roth | Cat#: 0335.3 | Final concentration: 4% (w/v) in PB or PBS |
| Chemical compound, drug | NBQX | Tocris Bioscience | Cat#: 1044 | Final concentrations: 200 nM and 50 μM |
| Chemical compound, drug | DAPI (1 mg/ml in water) | Thermo Fisher Scientific | Cat#: 62248 | Post hoc labeling (1:2000) |
| Software, algorithm | Python | Python Software Foundation | RRID:SCR_008394 | |
| Software, algorithm | NEST simulator | NEST Initiative | RRID:SCR_002963 | |
| Software, algorithm | Clampfit† | Molecular Devices | RRID:SCR_011323 | |
| Software, algorithm | Image J | NIH ‡ | RRID:SCR_003070 | |
| Software, algorithm | Prism (Graphpad) | GraphPad Software | RRID:SCR_002798 | |
| Other | OCT § | Method described in *Del Turco and Deller, 2007* | N/A | |

*Alexa Fluor 488-Conjugate.
†pClamp software package.
‡National Institutes of Health.
§Organotypic entorhinal-hippocampal tissue cultures.

## Preparation of tissue cultures

Entorhinal-hippocampal tissue cultures were prepared as published before (*Del Turco and Deller, 2007*; *Lenz et al., 2020*). All tissue cultures were cultivated for at least 18 days inside the incubator with a humidified atmosphere (5% $CO_2$ at 35 °C to reach an equilibrium status). The incubation medium consists of 50% (v/v) 1× minimum essential media (#21575-022, Thermo Fisher, USA), 25% (v/v) 1 × basal medium eagle (#41010-026, Thermo Fisher, USA), 25% (v/v) heat-inactivated normal horse serum, 25 mM HEPES buffer solution (#15630-056, Gibco), 0.15% (w/v) sodium bicarbonate (#25080-060, Gibco), 0.65% (w/v) glucose (#RNBK3082, Sigma), 0.1mg/ml streptomycin, 100U/ml penicillin, and 2mM glutamax (#35050-061, Gibco). The incubation, pre-warmed to 35 °C and adjusted to pH=7.38, as renewed three times per week throughout all experiments.

## Experimental protocol design

The primary objective of this study was to determine whether dendritic spine density is regulated in a monotonic or non-monotonic, dose-dependent manner, with the aim of developing a synapse-number-based structural plasticity rule that reflects biological reality. To this end, the competitive AMPA receptor antagonist NBQX (2,3-dioxo-6-nitro-7-sulfamoyl-benzofquinoxaline) was applied at varying concentrations to wild-type hippocampal slice cultures while recording synaptic transmission in CA1 pyramidal neurons. Two distinct levels of inhibition—'partial' and 'complete'—were defined based on the reductions in amplitude and frequency of sEPSCs. These same concentration ranges

were subsequently applied to Thy1-eGFP cultures for three days. Individual dendritic segments of CA1 pyramidal neurons were imaged before and after treatment to assess whether the two levels of AMPA receptor inhibition led to differential changes in spine densities and spine sizes.

## Whole-cell patch-clamp recordings

To assess the effects of different NBQX concentrations on synaptic transmission, whole-cell patch-clamp recordings were performed in CA1 pyramidal neurons at 35 °C. The bath solution contained (in mM) 126 NaCl, 2.5 KCl, 26 NaHCO$_3$, 1.25 NaH$_2$PO$_4$, 2 CaCl$_2$, 2 MgCl$_2$, and 10 glucose (aCSF) and was continuously oxygenated with carbogen (5% CO$_2$ / 95% O$_2$). Glass patch pipettes had a tip resistance of 4-6MΩ, filled with the internal solution which contained (in mM) 126 K-gluconate, 10 HEPES, 4 KCl, 4 ATP-Mg, 0.3 GTP-Na$_2$, 10 PO-Creatine, 0.3% (w/v) biocytin. The internal solution was adjusted to pH=7.25 with KOH and reached 290 mOsm with sucrose. sEPSCs were recorded from six CA1 pyramidal neurons per culture using whole-cell voltage-clamp at a holding potential of -70 mV. Series resistance was monitored before and after each recording, and neurons were excluded if it changed significantly or exceeded 30 MΩ. Each recording lasted 2 min.

## NBQX treatment

NBQX (2,3-dioxo-6-nitro-7-sulfamoyl-benzo[f]quinoxaline; at. No. 1044, Tocris Bioscience, Germany) is a competitive AMPA receptors antagonist (*Mathiesen et al., 1998*). To achieve partial or complete inhibition of AMPA receptor currents, we selected two concentrations——200 nM and 50 µM——delivered via bath application. Wild-type cultures were recorded in either standard ACSF or ACSF supplemented with NBQX at the indicated concentrations. For time-lapse imaging, Thy1-eGFP cultures were treated with 200 nM and 50 µM by adding NBQX to the incubation medium for three days. Fresh NBQX was added with each medium exchange.

## Tissue fixation and immunohistochemical staining

Recorded cultures were fixed and *post hoc* stained. Cultures were fixed by immersing into 4% (w/v) paraformaldehyde (PFA) in 1 × phosphate-buffered saline (PBS, 0.1M, pH=7.38) for 1 h and transferred into 1 × PBS for storage at 4 °C after being washed in 1 × PBS. Before staining, all fixed cultures were again washed three times with 1 × PBS (3 × 10 min) to remove residual PFA. We incubated the cultures with Streptavidin Alexa Fluor 488 Conjugate (1:1000, #S32354, Invitrogen, Thermo Fisher, USA) in 1 × PBS with 10% (v/v) in normal goat serum and 0.05% (v/v) Triton X-100 at 4 °C overnight. The following morning, cultures were rinsed with 1 × PBS (3 × 10 min) and incubated with DAPI (1:2000) in 1 × PBS for 20 min. After four additional washes in 1 × PBS (4 × 10 min), cultures were mounted on glass slides with DAKO anti-fading mounting medium (#S302380-2, Agilent) for confocal imaging.

## Time-lapse imaging

To assess whether dendritic spine densities and sizes were altered by the three-day NBQX administration, we performed time-lapse imaging of the same apical dendritic segments of CA1 pyramidal neurons before and after treatment. Live-cell imaging was conducted at a Zeiss LSM800 microscope equipped with 10 × (W N-Achroplan 10×/0.3 M27; 420947-9900-000, Carl Zeiss) and 63 × (W Plan-Apochromat 63×/1,0 M27; 421480-9900-000, Carl Zeiss) water-immersion objectives. Thy1-eGFP tissue cultures with clearly identified CA1 pyramidal neurons were selected, and dendritic segments from the stratum radiatum were imaged. During each imaging session, a membrane insert holding four cultures was placed in a 35 mm petri dish containing 5 ml of pre-warmed, pH-adjusted incubation medium maintained at 35 °C. Imaging was repeated after three days of treatment using the same dendritic segments. Three independent experimental batches were performed to ensure sufficient sampling for morphological analysis. In each batch, two to four cultures were included per group: vehicle-only control, 200 nM NBQX, and 50 µM NBQX treatment.

## Experimental data quantification

sEPSCs analysis used the automated event detection tool from the pClamp11 software package as previously described (*Lenz et al., 2021*). All experimental data were analyzed in a blinded manner to prevent bias during data interpretation.

Spine density quantification was performed in $z$-stacked fluorescent images of Thy1-eGFP cultures, which were projected to create a $2D$ representation of individual dendritic segments. ImageJ plugin Spine Density Counter (*Omedalus, 2022*) was used to count spine numbers and measure segment length, which estimates spine density. For the same dendritic segments imaged at different time points, special attention was paid to ensure the same starting and ending points were used. *Post hoc* visual inspection was applied to ensure the spine detection results were not strongly biased. In each image, one to five segments were selected for spine density, varying with the number of branches in the image. Both the raw values and normalized values by baseline spine density were used in the analysis. Statistical methods were specified in the individual results section. All spine density data were quantified in a blinded manner.

Spine size quantification was conducted in the same $z$-projected fluorescent images as spine density analysis by tracking individual spines. To eliminate the bias from drawing and automatic reconstruction, we drew circles manually around the spine to cut it from the dendrite; the spine size was estimated by measuring the signal intensity with an arbitrary unit of the drawn circle. The drawing and measurements were performed with FIJI ImageJ. We analyzed the spine sizes in each segment that has been analyzed for spine density. We followed 20 to 50 spines throughout the segment, depending on the length of the segment. Both the raw values and normalized values by baseline spine size were used in the analysis. Statistical methods were specified in the individual results section. All spine size data were analyzed in a blinded manner.

Data normalization was performed on the spine density and spine size data in the present study after analyzing raw data. For spine density, we calculated the ratio between the final density and the corresponding baseline density of the same dendritic segment. Spine size normalization followed the same protocol that we calculated the ratio between the final size and the corresponding baseline density of the same dendritic spine. In some scenarios, we also normalized the alteration of spine sizes $\Delta$ spine size by their baseline size before treatment. To do this, we first fetched the spine sizes of individual spines pre- and post-treatment to calculate the alteration ($\Delta$ spine size). Then, we normalized the amount of alteration by dividing it with the corresponding baseline spine size.

Data normalization and grouping was originally invented for this study (*Figure 8F*). After obtaining the normalized alteration of the spine sizes ($\Delta$ spine size), we binned the data by their initial sizes. By sorting the spine ID in each binned group, we grouped the spine size alterations ($\Delta$ spine size) by their initial spine size. This analysis shows us how spines with different initial sizes update their size over a certain time course with and without NBQX treatment.

## Statistical analysis

Dunn's multiple comparison test was applied for statistical analysis regarding the sEPSC events among the three groups. For spine density analysis, the Wilcoxon test was applied to compare the values of each segment before and after the three-day treatment. If not otherwise stated, 'ns' means not significant, '*' means $p < 0.05$, '**' means $p < 0.01$, '***' means $p < 0.001$. For spine size analysis, we first applied the Wilcoxon test for individual spines. Sample sizes were based on previous studies on structural and functional synaptic plasticity using similar slice culture preparations and were not determined by formal power analysis. Linear mixed model (LMM) was applied to double confirm the statistical significance by accounting for the data clustering within each segment. We applied LMM analysis *spine_size ~ timing* to compare the values before and after treatment with the segment ID as group factor, as used in *Lu et al., 2022*. The significance of the LMM results was judged by whether the confidence interval (CI) crossed zero. No data points were excluded as outliers; all replicates are reported. Samples were allocated to experimental conditions based on treatment type. No randomization was used.

## Neuron model and network model

We used the same spiking neuron model and network architecture as described before in *Gallinaro and Rotter, 2018* and in *Lu et al., 2019*; *Lu et al., 2022*. Current-based leaky integrated-and-fire point neuron was used for both excitatory and inhibitory neurons. We build an inhibition-dominated network with 10000 excitatory neurons and 2500 inhibitory neurons (*Brunel, 2000*). To simplify the scenario, we only grow the connections within the excitatory population (E-E) with the activity-based structural plasticity rule (see the Structural plasticity rule section below). Each inhibitory neuron was

**Table 1.** Parameters for the point neuron model.

| $\tau_m$ | $t_{\text{ref}}$ | $V_0$ | $V_{\text{reset}}$ | $V_{\text{th}}$ | $C_{\text{mem}}$ |
|---|---|---|---|---|---|
| 20.0 ms | 2.0 ms | 0.0 mV | 10.0 mV | 20.0 mV | 250 pF |

beforehand hard-wired randomly to receive synapses from 10% of the excitatory and inhibitory population. All details and parameters concerning neural and network models can be found in *Tables 1–7*. We performed network simulations with the simulation software NEST (NEural Simulation Tool) 2.20.2 and NEST 3.0 (*Fardet et al., 2021*) and MPI-based parallel computation.

## Structural plasticity rule

We enabled the growth, retraction, and rewiring of synapses among excitatory neurons with the help of structural plasticity rules. By definition, each excitatory neuron has multiple dendritic spines and axonal boutons, which are called synaptic elements. Synapses were formed by randomly matching free compatible synaptic elements. The growth and retraction of synaptic elements, or in other words, the number of synaptic elements, is governed by a growth rule. Three structural plasticity rules were explored in the current study: (i) linear growth rule, (ii) Gaussian growth rule with a zero setpoint and a non-zero setpoint, (iii) Gaussian growth rule with two non-zero setpoints. All three rules are determined by a function of calcium concentration that reflects neural activity,

$$\frac{\mathrm{d}}{\mathrm{d}t}C(t) = -\frac{1}{\tau_{\text{Ca}}}C(t) + \beta_{\text{Ca}}S(t), \tag{1}$$

where $C(t)$ is the time evolution of calcium concentration. Calcium concentration decays with a time constant $\tau_{\text{Ca}}$ and increases with calcium influx ($\beta_{\text{Ca}}$) upon the emission of an action potential $S(t)$ of the postsynaptic neuron. This operation is performed internally by the NEST simulator and works as a low-pass filtered signal of the spiking activity of the neuron. The growth of synaptic elements is regulated differently depending on the calcium concentration and the shape of the growth rule.

### Linear growth rule

The linear rule was first implemented in NEST in *Diaz Pier et al., 2016* and systematically studied in an inhibitory-dominant neural network (*Gallinaro and Rotter, 2018*; *Lu et al., 2019*; *Gallinaro et al., 2022*; *Lu et al., 2022*). The number of synaptic elements ($z(t)$) is linearly dependent on the calcium concentration,

$$\frac{\mathrm{d}}{\mathrm{d}t}z(t) = \nu\left[1 - \frac{1}{\epsilon}C(t)\right], \tag{2}$$

where $\nu$ is the growth rate, and $\epsilon$ is the target level of calcium concentration. Since calcium concentration reflects the neural activity loyally, the target level also suggests a setpoint of firing rate in the context of a certain neural network. As discussed before, when neurons fire below their target rate, they grow new synaptic elements and form new synapses. On the other hand, they break existing synapses and retract synaptic elements when they fire above the target rate (setpoint).

### Gaussian growth rule

The Gaussian rule has a more complex dependency on the calcium concentration when neural activity is too low, as shown in *Equation 3*. This rule was suggested and explored initially in *Butz and van Ooyen, 2013*.

**Table 2.** Parameters for the network model.

| $N_{\text{E}}$ | $N_{\text{I}}$ | $\Gamma_{\text{E}-\text{I}}$ | $\Gamma_{\text{I}-\text{E}}$ | $\Gamma_{\text{I}-\text{I}}$ | $J_{\text{E}}$ | $J_{\text{I}}$ | $r_{\text{ext}}$ |
|---|---|---|---|---|---|---|---|
| 10000 | 2500 | 10% | 10% | 10% | 0.1 mV | −0.8 mV | 30 kHz |

Varied values for $r_{\text{ext}}$ were used for activity perturbation.

**Table 3.** Parameters for the linear structural plasticity model.

| $\epsilon$ | $\nu$ | $\tau_{Ca}$ | $\beta_{Ca}$ |
|---|---|---|---|
| 0.0079 | 0.00395 s$^{-1}$ | 10 s | 0.0001 |

$$\frac{\mathrm{d}}{\mathrm{dt}}z(t) = \nu\left(2e^{-\left(\frac{C(t)-\xi}{\zeta}\right)} - 1\right),$$
(3)

where $\xi = \frac{\eta + \epsilon}{2}$, and $\zeta = \frac{\eta - \epsilon}{2\sqrt{\ln(1/2)}}$. In this rule, $\eta$ and $\epsilon$ are two setpoints: $\epsilon$ is the stable setpoint as used in the linear rule. When the neuron fires above $\epsilon$, it retracts synaptic elements as in the linear rule.

$\eta$ is another setpoint introduced specifically for the Gaussian rule, determining the regulation manner when the neuron activity drops below $\epsilon$. When the neuron is firing below $\epsilon$ but above $\eta$, the number of synaptic elements present will undergo homeostatic outgrowth, but when the neuron is firing below $\eta$, neurons will break synapses and retract elements. In the case where $\eta = 0$, neurons cease to change synaptic elements when their firing rate drops to zero.

### Homeostatic synaptic scaling

In order to achieve homeostatic synaptic scaling, we implemented a new synaptic model in NEST called *scaling_synapse*. In this synapse model, the weight of the synapse is regulated by the difference between a homeostatic setpoint and the calcium trace of the postsynaptic neuron,

$$\frac{\mathrm{d}}{\mathrm{dt}}w(t) = \rho w(t)(C(t) - \epsilon),$$
(4)

where $\rho$ is the scaling factor, and $\epsilon$ is the same target value as used in the HSP rule.

### Establishing a comparable neural network with the biphasic Gaussian rule

Although network development is not the primary focus of this study, we aimed to establish a network with statistics comparable to those obtained using the linear and zero Gaussian rules. This was necessary to enable systematic simulations with the biphasic Gaussian rule, which is supported by our experimental findings and constitutes the main focus of the present work. One solution was inspired by previous observation that increased neural excitability facilitates circuit development (*Johnson-Venkatesh et al., 2015*). We enhanced the excitability of excitatory neurons during network growth by applying a damping facilitating current ($I_{\text{facilitating}}$) to raise the average membrane potential closer to the threshold potential (*Figure 4D*). With this intervention, the neural network governed by the biphasic Gaussian rule successfully grew to the equilibrium state and maintained the dynamics after the facilitating current decayed to zero (*Figure 4E*). Given that varying the initial intensity of the facilitating current resulted in different equilibrium connectivity (*Figure 4F*), we used the intensity (750 pA) that produced the same network connectivity (10%) and firing dynamics as the linear rule for further exploration.

### Activity perturbation

To examine different activity-induced scenarios on neural network connectivity, we performed systematic activity manipulation to a subnetwork of excitatory neurons ($N_{sub}$=1000), by changing its Poissonian input from 0% to 200% fold of the original intensity (FOI). All manipulations were performed at 6000 s when the network had grown to the equilibrium state, respectively, with three structural plasticity rules. For the Gaussian rule with two non-zero setpoints, we applied damping current injection

**Table 4.** Parameters for the zero Gaussian structural plasticity model.

| $\epsilon$ | $\eta$ | $\nu$ | $\tau_{Ca}$ | $\beta_{Ca}$ |
|---|---|---|---|---|
| 0.0079 | 0 | 0.004 s$^{-1}$ | 10 s | 0.0001 |

**Table 5.** Parameters for the biphasic Gaussian structural plasticity model.

| $\epsilon$ | $\eta$ | $\nu$ | $\tau_{Ca}$ | $\beta_{Ca}$ |
|---|---|---|---|---|
| 0.0079 | 0.0007 | 0.004 s$^{-1}$ | 10 s | 0.0001 |

to the soma to facilitate its growth within the first 4000 s and the network was simulated for another 2000 s without any facilitating current.

## Quantifying firing rate, network connectivity, and synapse number

### Firing rate

Firing rate was calculated by the average spike count over a recording period for individual neurons. We have long intervals (5 s) and short intervals (1 s). Short intervals were only used within the short time window after activity perturbation to reveal its transient dynamics; otherwise, long intervals were used.

### Network connectivity

Two types of connectivity were used in the present study. We used a $N \times N$ connectivity matrix ($A_{ij}$) to represent the recurrent excitatory connections of our network, where columns and rows correspond to pre- and postsynaptic neurons. For structural connectivity, the entry $A_{ij}$ of the matrix represents the total number of synaptic connections from neuron $j$ to neuron $i$. For effective connectivity, we integrated the synapse number with individual weights for each pair of neurons by summating the total weights. So the entry of the connectivity matrix is the equivalent number of unit synapses, by dividing the sum with a uniform weight 0.1 mV. To average the mean connectivity of the whole network or a subnetwork at any given time $t$, corresponding columns and rows of the connectivity matrix were selected and averaged by $\Gamma(t) = \frac{1}{m\tilde{n}} \sum_{mn} A_{ij}$. Synapse numbers were calculated by the sum of the entry in the structural connectivity matrix.

### Synapse number

Input and output synapse numbers, also called indegree and outdegree in another context, were calculated by summating the input and output synapse numbers of individual excitatory neurons based on the structural connectivity matrix.

### Quantifying the discrepancies in firing rate and connectivity from the target values

To apply a systematic comparison among different combinations of the synaptic scaling strengths and structural plasticity growth rates, we summated the discrepancies in firing rate and connectivity from the target values for the subpopulation over time as an index of activity and connectivity recovery. All the discrepancies were calculated by estimating the area between the actual time course and the equilibrium connection probability (10%) or the target rate ($\epsilon$) from the time of silencing until the end of the simulation. The method was explained in detail in *Figure 9—figure supplement 3*. To achieve the heat maps in *Figure 9*, we averaged 11 random trials for each parameter combination.

## Acknowledgements

The work was supported by Deutsche Forschungsgemeinschaft (DFG; Project-ID 259373024 B14–CRC/TRR 167 to AV). The research leading to these results has received funding from the European Union's Horizon 2020 Framework Programme for Research and Innovation under the Specific Grant Agreement No. 945539 (Human Brain Project SGA3). This research has also been partially funded by the Helmholtz Association through the Helmholtz Portfolio Theme Supercomputing and Modeling

**Table 6.** Parameters for the synaptic scaling model.

| $\epsilon$ | $\rho$ |
|---|---|
| 0.0079 | [0, 0.1] |

**Table 7.** Protocols for numeric stimulation.

| | plasticity rules | $\eta$ | growth rate | $I_{\text{facilitating}}$ | FOI | $\tau_{\text{Ca}}$ | $\rho$ |
|---|---|---|---|---|---|---|---|
| *Figure 3Bi, Di* | linear | - | 100%, 50%, 10% | | | 10 s | - |
| *Figure 3Bii, Dii* | Gaussian | 0 | 100%, 50%, 10% | | | 10 s | - |
| *Figure 3Biii, Diii* | Gaussian | 0.0007 | 100%, 50% | | | 10 s | - |
| *Figure 4A-C* | Gaussian | 0.0007 | 100% | 750 pA | | 10 s | |
| *Figure 4E* | Gaussian | 0.0007 | 100% | 750 pA | | 10 s | - |
| *Figure 4F* | Gaussian | 0.0007 | 100% | 0, 200, 500, 750 pA | | 10 s | - |
| *Figure 5B and C* | Gaussian | 0.0007 | 100% | 750 pA | 110%,95%, 0% | 10 s | - |
| *Figure 5D* | Gaussian | 0.0007 | 100% | 750 pA | [0%, 200%] | 10 s | - |
| *Figure 5E* | - | - | - | - | - | 1 s, 10 s | - |
| *Figure 5F*, orange | Gaussian | 0.0007 | 100% | 750 pA | 0% | 1 s | - |
| *Figure 5F*, dark blue | Gaussian | 0.0007 | 100% | 750 pA | 0% | 10 s | - |
| *Figure 5F*, light blue | Gaussian | 0.0007 | 10% | 750 pA | 0% | 10 s | - |
| *Figure 6B and C* | Gaussian | 0.0007 for D,0.0004 or0.001 for A | 10% | 750 pA | 0% | 10 s | - |
| *Figure 6E and F* | Gaussian | 0.0007 | 10%, 100% | 750 pA | 0% to100% or0% to200% | 10 s | - |
| *Figure 7B–F* | Gaussian +scaling | 0.0007 | 10% | 750 pA | 0% | 10 s | 0, 0.01, 0.02 |
| *Figure 8C–F* | Gaussian +scaling | 0.0007 | [0%, 100%] | 750 pA | 0% | 10 s | [0, 0.1] |

The column growth rate $\nu$ summarizes the percentages of the original growth rate used for structural plasticity.

The column FOI summarizes the percentages of original intensities used for external activity perturbation.

For *Figure 5D*, FOI values ranging from 0% to 200% were used for the systematic study.

*Figure 5E* was performed in a single neuron model without any plasticity.

In *Figure 6BC*, distinct $\eta$ values were used for axonal and dendritic elements. Otherwise, the same value was used for both parts.

In *Figure 6E and F*, the silencing protocol was applied (0% FOI) and then external stimulation was applied with 100% or 200% FOI.

In *Figure 7B–F*, only three values for scaling factor ($\rho$) were used.

In *Figure 8C–F*, a series values for $\rho$ ranging from 0 to 0.1 and a series of values for growth rate ranging from 0% $\nu$ to 100% $\nu$ were used for the systematic study.

for the Human Brain. We acknowledge the use of Fenix Infrastructure resources, which are partially funded by the European Union's Horizon 2020 research and innovation program through the ICEI project under grant agreement No. 800858.

## Additional information

### Funding

| Funder | Grant reference number | Author |
|---|---|---|
| Deutsche Forschungsgemeinschaft | Project-ID 259373024 B14-CRC/TRR 167 | Andreas Vlachos |
| European Union | Horizon 2020 Framework Programme under the grant agreement No. 945539 | Sandra Diaz-Pier |
| European Union | Horizon 2020 Framework Programme under the grant agreement No.800858 | Sandra Diaz-Pier |
| Helmholtz Association | Helmholtz Portfolio Theme "Supercomputing and Modeling for the Human Brain" | Sandra Diaz-Pier |

The funders had no role in study design, data collection and interpretation, or the decision to submit the work for publication.

### Author contributions

Han Lu, Conceptualization, Data curation, Formal analysis, Investigation, Visualization, Methodology, Writing – original draft, Writing – review and editing; Sandra Diaz-Pier, Software, Formal analysis, Validation, Methodology, Writing – review and editing; Maximilian Lenz, Data curation, Formal analysis, Validation, Investigation, Methodology, Writing – review and editing; Andreas Vlachos, Conceptualization, Supervision, Funding acquisition, Methodology, Project administration, Writing – review and editing

### Author ORCIDs

Han Lu ⓘ http://orcid.org/0000-0002-3508-2208
Sandra Diaz-Pier ⓘ http://orcid.org/0000-0002-3168-5394
Maximilian Lenz ⓘ http://orcid.org/0000-0003-3147-4949
Andreas Vlachos ⓘ https://orcid.org/0000-0002-2646-3770

### Ethics

All animal experiments were approved by the appropriate animal welfare committee and the animal welfare officer of Albert-Ludwigs-University Freiburg, Faculty of Medicine under X-21/01B and X-18/02C. All effort was made to reduce the pain or distress of animals.

Reviewer #1 (Public review): https://doi.org/10.7554/eLife.88376.3.sa1
Reviewer #2 (Public review): https://doi.org/10.7554/eLife.88376.3.sa2
Author response https://doi.org/10.7554/eLife.88376.3.sa3

## Additional files

### Supplementary files

MDAR checklist

## Data availability

All raw data, simulation code, and analysis scripts supporting this study are publicly available at: https://github.com/ErbB4/HSP-SS-interplay (copy archived at *Lu, 2024*).

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
