## [Editor Report · eLife Assessment]

This **valuable** study combines experiments and modelling to advance our understanding of the nonlinear nature of homeostatic structural plasticity and its interaction with synaptic scaling. The methodology and findings are **solid**, although additional work is needed to better link models with experiments and support some of the conclusions drawn. This study will be of interest to theoretical and experimental neuroscientists working in homeostatic plasticity.

---

## [Referee Report · Reviewer #1 (Public review)]

This manuscript investigates homeostatic structural plasticity and its interplay with synaptic scaling. It uses an integrated approach with models and experiments.

First, electrophysiology and chronic imaging are used to investigate the influence of different levels of AMPA-receptor antagonist NBQX, which allows for gradual activity reduction. Low levels of NBQX lead to a decrease of activity and a homeostatic increase of synapse density, whereas high levels block neural activity and lead to a reduced number of synapses after 3 days. The authors conclude that there must be a non-linear dependency between neuronal activities and rewiring. As a mathematical model for this, a biphasic structural plasticity rule is used, which, for increasing neural activities, switches from net synapse removal to growth and back, yielding two stable states at zero activity and the homeostatic target.

This rule is tested in various situations in silico, yet without attempting to reproduce the experiment. First, in network development, the biphasic rule generates a lot of unconnected silent neurons and a reasonable network structure only emerges when the neurons are additionally supported by a facilitating input current. For comparison, a linear and a simpler nonlinear homeostatic plasticity model, which had been ruled out by the experimental data, need no external drive. Second, the consequences of lasting, altered stimulation in a subgroup of neurons is explored. As expected by the design of the rule, a small increase and decrease in stimulation leads to a decrease and increase of synaptic connectivity, respectively, and stimulation silencing led to a complete disconnection of the sub-population with restoration of activity. Unlike in previous studies, an asymmetry of pre- and postsynaptic plasticity mechanisms cannot rescue this. Third, silencing only for a short time period and then overstimulating the network led to overly strong activity, which may, however, also hold without silencing. For a transiently silenced stimulation, recovery is possible, but only when there is enough recurrent excitation from the rest of the network.

Following this, the second part of the manuscript explores whether synaptic scaling may adapt and up-regulate the recurrent excitation, such that activity in a normally silenced subpopulation can be restored. Indeed, fast enough synaptic scaling leads to a recovery of neuronal activity in simulations, but leads to highly synchronous activity. A systematic model analysis shows at which scaling and rewiring speeds the activity and connectivity for a silenced sub-population can be restored. In between, however, the authors analyze spine sizes and changes in their whole population AMPAR-blocking experiments that demonstrate synaptic scaling and that structural plasticity and scaling effects may be jointly regulated. This experimental "break" between a simulation and its systematic analysis makes the paper harder to read and seems unnecessary as the analyses from the experiments are not repeated for the model.

Overall, the combination of experiments and simulations is a promising approach to investigate network self-organization. Especially the gradual blocking of activity is very valuable to inform mathematical models and distinguish them from alternatives. However, it remains unclear whether the model would actually reproduce the experiment. When switching from one to the other, this entails a detour to the conceptual level which makes the narrative sometimes hard to follow.

In summary, this manuscript makes a valuable contribution to discern the mathematical shape of a homeostatic structural plasticity model and understanding the necessity of synaptic scaling in the same network. Both experimental and computational methods are solid and well described. Yet, both parts could be linked better in order to obtain conclusions with more impact and generality.

---

## [Referee Report · Reviewer #2 (Public review)]

This manuscript by Lu et al addresses the understudied interplay between structural and functional changes underlying homeostatic plasticity. Using hippocampal organotypic slice cultures allowing chronic imaging of dendritic spines, the authors showed that a partial or complete inhibition of AMPA-type glutamate receptors differentially affects spine density, respectively leading to an increase or decrease. Based on that dataset, they built a model where activity-dependent synapse formation is regulated by a biphasic rule and tested it in stimulation- or deprivation-induced homeostatic plasticity. The model matches experimental data (from the authors and the literature) quite well, and provides a framework within which functional and structural changes coexist to regulate firing rate homeostasis.

While the correlation between changes in AMPAR numbers and in spine number/size has been well characterized during Hebbian plasticity, the situation is much less clear in homeostatic plasticity due to multiple studies yielding diverging results. This manuscript adds new experimental results to the existing data and presents a valuable effort to generate a model that can explain these divergences in a unifying framework.

The model and its successive implantation steps are well presented along a clear thread. However, the manuscript would benefit from clarifications at several key points (Hebbian vs homeostatic timeline).

First of all, it would have benefited from having an actual timeline of structural changes throughout the three days of AMPAR inhibition, especially as their experimental model allows it. This would have provided much-needed and otherwise entirely lacking information on spine dynamics (especially on transient spines) and on the respective timescale of the structural and functional changes, instead of modelling an entire timeline based solely on an experimental endpoint.

Additionally, the model would have been strengthened by an experimental dataset with homeostatic plasticity induced by higher activity (e.g. with bicuculline). To the best of my knowledge, there is currently no data on structural plasticity following scaling down, and it is also known that scaling up and down are mediated by different molecular pathways. The extension of the model from scaling up (in response to silencing) to scaling down (in response to increased activity) offers an interesting perspective, but its biological relevance is limited as there is no experimental data to support it.

Finally, the difference between weak and complete inhibition could have been more extensively characterized. The authors focus indeed on the effects of either condition on spine number, but only integrate synaptic weights following complete inhibition. This is a pity, as they show some intriguing data suggesting a differential effect on spine size by partial or complete AMPAR inhibition (although further work is required to support some of their interpretations). Since the model aims at correlating structural and functional homeostatic plasticity, the fact that it is only demonstrated for one of the two conditions tested severely undermines the claims of the authors in the discussion that the model tackles that question.

---

## [Author Response]

The following is the authors’ response to the original reviews.

**Reviewer #1 (Recommendations For The Authors):**
(1) Gap of knowledge:From the introduction, I got the impression that the manuscript tries to answer the question of whether homeostatic structural plasticity is functionally redundant to synaptic scaling. However, the importance of this question needs to be worked out better. Also, I think it is hard to tackle this question with the shown experiments as one would have to block all other redundant mechanisms and see whether HSP functionally replaces them.

We appreciate the reviewer’s valuable feedback regarding the relationship between homeostatic structural plasticity (HSP) and synaptic scaling. The main objective of our study is indeed to investigate whether structural plasticity is homeostatically regulated, and if so, whether it acts as a redundant or heterogeneous mechanism in relation to synaptic scaling, which is widely recognized as a primary homeostatic process.

In our revised introduction, we have clarified this central question and its significance. Specifically, we explored why experimentally observed changes in spine density, a measure of structural plasticity, do not exhibit the same homeostatic characteristics as changes in spine head size, which reflects synaptic scaling, particularly under conditions of activity blockade.

We hypothesized two key points:

(1) Structural plasticity may not follow a monotonically activity-dependent rule as strictly as synaptic scaling.

(2) The observed changes in spine density may be influenced by the simultaneous modulation of spine size, suggesting that structural plasticity and synaptic scaling interact within the same biological system.

Both hypotheses were tested through a combination of experimental observations and systematic computer simulations. Our conclusions demonstrate that spine-number-based structural plasticity follows a biphasic activity-dependent rule. While it largely overlaps with synaptic scaling under typical conditions, it exhibits heterogeneity under extreme conditions, such as activity silencing. Furthermore, our simulations revealed that both mechanisms can compete and complement each other within neural networks.

We believe that these results offer a nuanced understanding of the interaction between structural plasticity and synaptic scaling, highlighting their redundancy under most conditions but also their heterogeneity under specific circumstances. Blocking all other redundant mechanisms, as suggested, would provide a more reductionist view, which may not capture the complexity and interplay of these processes in a physiological setting. Our approach reflects this complexity, providing insight into how these mechanisms operate together in a naturalistic context.

We have revised the introduction to better convey these points and emphasize the significance of this question for understanding the dynamics of homeostatic regulation in neural networks.

Similarly, the simulations do not really tackle redundancy as, e.g. network growth cannot be achieved by scaling alone.

We appreciate the reviewer’s comment regarding synaptic scaling's limitations in achieving network growth. We would like to clarify that we did not intend to suggest that structural plasticity and synaptic scaling are fully redundant. In fact, it is well established in the literature that structural plasticity plays a dominant role during development, particularly in network growth, which synaptic scaling alone cannot achieve.

The primary objective of our study was to investigate the interaction between structural plasticity and synaptic scaling under conditions of activity perturbation, rather than during network growth or development. To avoid any confusion regarding developmental processes, we chose to grow the network using only structural plasticity in our simulations. Synaptic scaling was then introduced (or not) during the phase of activity deprivation to specifically examine its role in regulating homeostasis under these conditions.

We have revised the corresponding sections of the manuscript to clarify this distinction, and we have ensured that the simulations reflect our focus on activity perturbation rather than network development. This distinction should help readers avoid conflating developmental processes with the specific goals of our study.

Instead, the section on "Integral feedback mechanisms" (L112-129) contains a much better description of the actual goals of the paper than is given in the introduction. Moreover, this section does not seem to include any new results (at least the Ca-dependent structural plasticity and synaptic scaling rules seem to be very common for me). I, therefore, suggest fusing this paragraph in the introduction to obtain a clearer and better understandable gap of knowledge, which is addressed by the paper.

We agree that the "Integral feedback control" section provides key information relevant to both the Introduction and Methodology. It outlines the theoretical framework and serves as a basis for the experimental design.

To better reflect this, we have revised the Introduction to include the gap in knowledge. However, we opted to retain the section in the Results, slightly modified, to set the context for the first experiment.

Along this line, as it seems a central point of the manuscript to distinguish the controller dependencies on Calcium, the different dependencies (working models) should be described in more detail. Also, the description of the inconsistencies of the previous results on HSP can be moved from the discussion (l419-l441) to the introduction.

We have revised the manuscript to place less emphasis on the controller models while retaining the core principles of control theory. The description of the HSP model has been moved to the Introduction, as suggested, while the detailed history remains in the Discussion to maintain the manuscript's consistency.

Systematic text revision: Regarding comment (1), we thank the reviewer for suggesting the text reorganization. We have adjusted several parts in the introduction, M&M section, and results section to increase clarity.

(2) Pharmacological Choice:It should be discussed why NBQX is used to induce the homeostatic effect instead of TTX. As there are studies showing that it might block homeostatic rewiring (doi.org/10.1073/pnas.0501881102) as well as synaptic scaling (10.1523/JNEUROSCI.3753-08.2009), it seems unclear whether the observed effects are actually corresponding to those in other publications.

The rationale for using NBQX in our experiments, rather than TTX, is detailed in the public response. We selected NBQX based on specific experimental motivations relevant to our study’s objectives, while acknowledging the potential differences in effects compared to other studies.

Local text revision: We added one paragraph in the discussion section to explain the idea better.

(3) Model-Experiment Connection:The paper combines simulations with experimental work, which is very good. However, in my opinion, the only connection between the two parts is that the experiments suggest a non-monotonic dependency between firing rate and synapse density (i.e. the biphasic dependency). The rest of the experimental results seem to be neglected in the modeling part. It is not even shown that the model reproduces the experiments. Instead, the model is tested in different situations and paradigms (blocking AMPARs in the whole culture vs network growth or silencing a sub-population). I think it would make the paper stronger and more consequential when a reproduction of the experiment by the model is demonstrated (with analogue analyses).

The experimental results serve three main purposes. First, as the reviewer noted, the spine analysis was conducted to inform the biphasic rule. Second, spine size analysis was performed to replicate published findings and confirm our modeling results, showing that activity deprivation leads to fewer synapses with larger sizes or higher weights. Third, the correlation analysis of spine density and size across dendritic segments suggested a hybrid combination of two types of plasticity across different neurons.

While we addressed these aspects in the Results and Discussion sections, the collective presentation in Fig. 2 may have caused some confusion. To improve clarity, we have now split the experimental results, presenting them alongside the relevant modeling data in Fig. 2, Fig. 8, and Fig. 9.

Also, there are a few more mismatches between the experiment and the model that you will want to discuss:• The size-dependent homeostatic effect (l154ff, Fig2F) is not reflected by the used scaling model.

We revised Fig 8 and the corresponding text to explain how the scaling model reflects such an effect.

• The model assumes reduced Ca levels. Yet, the experimental protocol blocks AMPARs, which are to my knowledge not the primary source of Ca influx, but rather the NMDARs.

The model is based on neural activity, with calcium concentration serving as an internal integral signal of the firing rate, allowing for integral control. While calcium plays a critical role in homeostasis, we caution against drawing a strict correspondence between the model's calcium dynamics and the experimental protocol, as calcium can be sourced from multiple pathways in neurons beyond AMPARs, such as NMDARs, voltage gated calcium channels, and intracellular stores. Also, our recent work demonstrated that under baseline conditions, the majority of AMPARs are not Ca2+ permeable, i.e., GluA2-lacking (Kleidonas et al., 2023)

Improving the calcium dynamics, including secondary calcium release and calcium stores, is part of our future plan to refine the HSP model and address experimental findings that are not fully explained by the current model.

• The model further assumes silencing by input removal, whereas the recurrent connections stay intact. Wouldn't this rather correspond to a deafferentation experiment, where connections to another brain area are cut?

Thank you for pointing at this. The modeling section was not intended to directly replicate the tissue culture experiments but rather to provide insights into a broader range of scenarios, including pharmacological treatments, deafferentation, lesions, and even monocular deprivation.

Systematic text revision: Regarding comment (3), the goal of our modeling work was more than reproducing. To better serve the purposes of experimental results used in the present study, to inform, confirm, and inspire, we have systematically adjusted the layout of experimental and modeling results to link them better.

(4) Is the recurrent component too weak?Your results show that HSP does not restore activity after silencing (deafferentation), whereas you discuss that earlier models did achieve this by active neighbors in a spatially organized network. However, the silenced neurons in your simulations also receive inputs through the "recurrent" connections from their neighbors (at least shortly after silencing). Therefore, given the recurrent input is strong enough, they should be able to recover in a similar way as the spatially organized ones. As a consequence, I obtained the impression that, in your model networks, activity is strongly driven by external stimulation and less by recurrent connections. I understand that this is important to achieve silencing through removing the Poisson stimulation. Yet, this fact may be responsible for the failure to restore activity such that presented effects are only applicable for networks that are strongly driven by external inputs, but not for strongly recurrent networks, which would severely limit the generality of the results. As a consequence, the paper would benefit from a systematic analysis of the trade-off between recurrent strength and input strength. Maybe, different constant negative currents could be injected in all neurons, such that HSP creates more recurrent synapses in the network.

We appreciate this insight. However, increasing recurrent input strength is beyond the scope of the current study, as it would fundamentally alter the predefined network dynamics of the Brunel network used. As noted in the manuscript, complete isolation or cell death is not always the outcome after input deprivation, lesion, or stroke, which cannot be fully explained by the Gaussian HSP rule alone. Butz and colleagues offered a solution using growth rules that maximized recurrent input, and we recognize the importance of their work.

That said, we approached the issue from a different angle, emphasizing the role of synaptic scaling in recurrence rather than relying solely on recurrent input strength. In biological networks, external inputs may vary, recurrency can be weak or strong, and synaptic scaling can dominate. Our model offers a complementary hypothesis, suggesting that these factors, in combination, contribute to the diverse and sometimes contradictory results found in the literature, rather than posing a strict constraint on network topology.

Local text revision: We emphasized these points in the Discussion section again.

(5) Missing conclusions / experimental predictionsAs already described, the modelling work is not reproducing the presented or previous experimental data. Hence, the goal of modelling should be to derive a more general understanding and make experimental predictions. Yet, the conclusions in the discussion stay superficial and vague and there are no specific experimental predictions derived from the model results.For example, the authors report that the recovery of activity in silenced cultures is observed in a previously spatially structured model but not in theirs -- at least with slow or no scaling. Yet it is left to the reader to think about whether the current model is an improvement to the previous one, how they could be experimentally distinguished, or to which experimental findings they relate or compare, which I would expect at this point. I would advise reworking the discussion and thoroughly working out which new insights the modelling part of the study has generated (not to be confused with the assumptions of the model aka the biphasic plasticity rule) and relating them to experimental pre- and postdiction.

We recognize the reviewer’s concern, which is closely related to comment (4). We have addressed these points by reorganizing the text to better clarify the purpose of our experimental work and its connection to the modeling results.

Specifically, we have reworked the discussion to highlight the new insights gained from the modeling, and how these can inform experimental predictions and interpretations. This includes distinguishing our model from previous ones and providing clearer connections to experimental findings.

Systematic text revision: Most of the comments on combining experiments and modeling results and on developing the story based on our expectations raised here are sincere and may also reflect the expectations and concerns of a broader readership, so we have accordingly adjusted the text in the Results and Discussion sections to make our points clear.

Suggestions for minor changes:Fig 1I: Please check the graph and make it more self-explaining. For example, mark the "setpoint" activity (in my opinion, both curves should be at baseline there. In that case, however, I do not see the biphasic behavior anymore). Maybe the table and the graph can be aligned along the activity axis? Also: synaptic inhibition should be increased and not decreased, right?

Local text and figure revision: I guess the reviewer meant for Fig. 2I? We have improved the visualization to avoid confusion.

L74-81: I would reverse the order of associative and homeostatic plasticity in this paragraph.

Local text and figure revision: We have fine-tuned the order in the first and second paragraphs to match the readers' expectations.

L74-75: Provide references for such theories.

Local text and figure revision: fixed.

L84-86: Please provide a reference for the claim that negative feedback, redundancy, and heterogeneity contribute to robustness.

Local text and figure revision: fixed.

L 95-97: I think the heterogeneity aspect needs to be worked out a bit better. Do you mean that the described mechanisms contribute to firing rate homeostasis in a different mixture for each neuron (as shown assumed in the last figure)?

Local text and figure revision: The term heterogeneity is used in the manuscript for two major different settings: (1) heterogeneity in terms of control theory and (2) different combinations of HSP and SS rules. We have named the second condition as diversity to avoid confusion.

L 132: The question of linearity has not been posed so far. Also, I think "monotonous" would be a much better term than linear (as a test for linearity would require more than 2 datapoints).

Local text and figure revision: We agreed linear is not a good term. We replaced it with ‘monotonic’ throughout the manuscript.

Fig2 Bii: The data for 50um is clearly not Gaussian.

We did not imply that the 50 µM condition is Gaussian. Instead, we noted that the non-linearity observed in both the 200 nM and 50 µM data suggests a non-monotonic growth rule rather than a linear one. We applied the Gaussian rule because it has been extensively studied in previous simulations, allowing us to benchmark our findings against those results.

Fig2 D, E inset: The point at time 0 does not convey any information and could be left out.

The time zero data is included to demonstrate that the three groups have a similar baseline, ensuring that any observed differences are due to the treatment and not pre-existing biases in the grouping.

L 178: As the Gaussian rule drops below zero above the upper set-point again, it is rather tri-phasic than bi-phasic.

We intended to convey that inhibition results in either spine growth or deletion, reflecting a bi-phasic response rather than a true tri-phasic one.

Fig 6A: You may want to mark the eta variables in the curves.

Local text and figure revision: fixed.

Fig 6E: The curve of the S population extending to the next panel looks a bit messy.

We retained the curve extension to visually convey the impression of excessive network activity.

L272: It needs to be better described/motivated how protocol 1 and 2 are supposed to study the role of recurrent connection as well as what kind of biological situation this may be.

Local text and figure revision: The corresponding text has been adjusted to avoid confusion.

L 272: It is not clear how faster simulation leads to less recurrent connectivity, when the stimulation protocol and the rates stay the same and the algorithm compensates for the timestep properly. Maybe you rather want to say that you silence 10x longer and stimulate 10x longer?

Local text revision: The corresponding text has been adjusted to avoid confusion.

L. 302: "reactivate"?

Local text revision: fixed.

L 322f: I would suggest showing the connectivity matrix for a time-point with restored activity as well.

Local text and figure revision: fixed.

Fig 8A: The use of the morphological reconstructions is a bit misleading as the model uses point neuron.

Local text revision: Now after reorganization, it is in Fig.9. We kept the reconstruction figure for motivational purposes, suggesting how to understand the meaning of the combinations in more biologically realistic scenarios. The corresponding text has been adjusted to avoid confusion.

Fig 8E-F: the y axis should be in the same orientation as in panel D.

Local text and figure revision: Good idea and fixed in the new Fig. 9.

Fig. 8F: The results here look a little bit random. Maybe more runs with the same parameters would smooth out the contours or reveal a phase transition.

Local text and figure revision: Thank you for the suggestion. We conducted an additional ten random trials to average the traces and heatmaps, improving the clarity of the results now presented in Fig. 9.

L411: Note that there are earlier HSP models by Damasch and van Ooyen & van Pelt, that might be worth discussing here.

Local text revision: fixed.

L416 "beyond synaptic scaling" reference needed.

Local text revision: fixed.

L419: The biphasic rule was suggested by Butz already.

Local text revision: We adjusted the text to emphasize our contribution in suggesting/confirming the biphasic rule based on direct experimental observations.

L 419-44: Most of this is actually state-of-the art and may be better placed in the introduction to justify the use of NBQX as a competititve blocker.

Local text revision: We adjusted the text in the introduction and Discussion sections to cover the raised points.

L487: In my opinion, although scaling adapts the weights quickly, the information about deviating firing rate is still stored in the calcium signal such that it will also give rise to structural changes (although they may be small when the rate is low). Thus, I think that fast scaling does not abolish structural changes.

Local text revision: We adjusted the text to account for other factors that could lead to the same or opposite conclusions.

L502f: Sentence unclear. Do you mean Ca is an integrated (low-pass filtered) version of the firing rate?

Yes.

L504: What is the cumulative temporal effect of error in estimating firing rates?

We were referring to the potential instability in numeric simulations if the firing rate is not tracked by an integral signal (calcium concentration) but is instead estimated through average spike counts over time. In our model, calcium serves as a proxy for the firing rate to guide homeostatic structural plasticity. The intake and decay constants are set to minimize the accumulation of errors over time, making long-term error accumulation unlikely. In any case, this is not intended to be a precise measure of the firing rate but rather a smooth guide for homeostatic control.

Local text revision: We rewrote the section so as not to cause extra concerns.

L505: Which two rules are meant here? Ca- and firing rate based or HSP and scaling?

Local text revision: The two rules are the HSP rule and the HSS rule. We have adjusted the text to improve clarity.

L505ff: I did not really understand the control theoretic view here and Supp Fig 5 is not self-explaining enough to help. In my view, scaling is a proportional controller for the calcium level (the setpoint is defined for calcium and not firing rate). Also, all of the HSP rules do neither contain an integral nor a differential of the error and are thus nonlinear but proportional controllers in first approximation. If this part is supposed to stay in the manuscript, the supporting information should contain a more detailed mathematical explanation. Relevant previous work on homeostatic control by synaptic scaling and homeostatic rewiring, e.g. doi: 10.23919/ECC54610.2021.9655157 should be discussed

Local text revision: We have updated the last paragraph to increase clarity. The HSP and HSS rules are proportional and integral for neural activity, as neural firing rate homeostasis is the meaningful goal. However, it is also correct that the integral component is gone if we view calcium concentration as the goal or setpoint. This paper is discussed and cited in a paragraph above this one.

**Reviewer #2 (Recommendations For The Authors):**
I have some additional suggestions and questions for the authors, which I am presenting following the order of the figures.Fig 1A: I'm a little bit puzzled by the timescales between Hebbian and homeostatic plasticity; a wealth of data suggests that Hebbian plasticity acts on a faster timescale than homeostatic plasticity, while Aii-Aiii implies the opposite. In lesion-induced degeneration, for instance, which is mentioned later by the authors, spine loss has been suggested to be Hebbian (LTD) while the subsequent recovery is homeostatic. Additionally, it will not be clear to the reader if the same stimulus could induce Hebbian and homeostatic plasticity, or why; the rest of the manuscript seems to imply that any stimulus could and would trigger homeostatic plasticity, which is not the case. Finally, there should be a mention somewhere that Hebbian structural plasticity also exists.

Local text and figure revision: We thank the reviewer for pointing out the time scale issue, which was not explicitly considered here and is now updated.

Fig. 2Bii: There is no significant difference at 200nm NBQX for sEPSC amplitude, contrary to what is stated in the text (line 136). Which one is it?

Local text revision: We thank the reviewer for pointing out the mistake. We have inspected the original statistical file and corrected the text.

Fig. 2F: The description of Fig. 2F in the text confused me for the longest time. I am still unsure why 200nm NBQX is described as leading to a general size increase when it follows the control line so closely, crosses 0 at the same point, and is even below the control line for the largest spine sizes. Similarly, 50um NBQX neatly overlaps with the control condition except for the smallest and largest spines, so the "shrinkage of middle-sized spines" doesn't seem different from the control condition. I also couldn't find any data supporting the statement that 50um NBQX increased only the size of "a small subset of large spines". Maybe the authors could clarify this section? I would also suggest adding statistics between the treatments at each spine size bin to support the claims, as they are central to the rest of the paper.Importantly, there is no description of the normalization nor the quantification of the difference between days in the methods; I am assuming post-pre for the difference and (post-pre)/pre for the normalization, but this should be much more detailed in the methodology. I was happy to see the baseline raw spine sizes in Supplementary Fig. 1, and would also suggest adding the raw spine sizes after treatment for comparison.

Local text and figure revision: We have adjusted the text and figure to improve clarity.

Fig. 2G/S2A: a scale for the label sizes would be helpful. I would also like to have the same correlation for 50um NBQX treatment and the control condition (at least in the supplementary figures).

Local text and figure revision: We have adjusted the text and figure to improve clarity.

Fig. 2I: I might be missing something, but why is the activity line flat when there are changes in spine density and size?

Local text and figure revision: We have adjusted the text and figure to improve clarity.

Fig. 3C-D: they are referenced in the text as Fig. 1C-D (lines 188-194).

Local text revision: fixed.

Fig. 5: it is interesting that the biphasic model captures both spine loss and recovery, fitting well with lesion-induced degeneration and recovery. Does this mean that the model captures other types of plasticity, or does it suggest to the authors that both steps are homeostatic?

Indeed, the biphasic HSP rule captures two types of activity dependence. The pioneering work by Gallinaro and Rotter (2018) also demonstrated that the HSP rule, even in its monotonic/linear form, exhibits associative properties, which are typically associated with Hebbian plasticity.

Fig. 6A: This figure requires a more detailed legend - what are the various insets? Does the top right graph only have one curve because they are overlapping and the growth rules are the same for axons and dendrites?

Local text revision: fixed.

Fig. 6E: There is usually an overshoot when a stimulus is removed, in this case at the end of the silencing period (as shown in Fig. 1Aiii). Is there a reason why this is not recapitulated here? It shouldn't be as extreme as in the right panel so there should be no degeneration.

We agree that removing the stimulus would typically trigger an opposite homeostatic process. However, in this protocol, we aimed to emphasize the role of recurrency by presenting extreme cases to illustrate potential scenarios for the readers.

Local text revision: We revised this paragraph to walk the readers through the rationale better.

Fig. 6: the authors mention distance-dependent connectivity (line 268), but I couldn't find any data related to that statement. I was particularly curious about that aspect, so I would like to know what this statement is based on, especially as they touch again on the role of morphology in Fig. 8, and distance-dependent connectivity is more prominent in the discussion. On a similar note, would the authors have data from other layers of CA1 that would show similar or other rules? Please note that I am not asking to include these data in the present paper - I am just curious if these data exist (or if the experiments are considered).

Such an extensive dataset is included and thoroughly investigated in another study that has just been published in Lenz et al., 2023. We updated the reference in the revised text.

Fig. 7E top: the scalebar is missing.

Local text revision: fixed.

Fig. 8A: do the colors have meaning? If yes, please state them. Also indicate that the left two neurons are pyramidal cells from CA1 and the right neurons are granule cells from the dentate gyrus.

Local text revision: fixed.

Line 302: "reactive" should be "reactivate".

Local text revision: fixed.